EFI-25-6
MPP-2025-107
LMU-ASC 13/25

# Quantum Calabi–Yau Black Holes
# and Non-Perturbative D0-brane Effects

**Alberto Castellano**[1,2], **Dieter Lüst**[3,4], **Carmine Montella**[3], **Matteo Zatti**[3]

[1]*Enrico Fermi Institute & Kadanoff Center for Theoretical Physics,*
*University of Chicago, Chicago, IL 60637, USA*

[2]*Kavli Institute for Cosmological Physics,*
*University of Chicago, Chicago, IL 60637, USA*

[3]*Max-Planck-Institut für Physik (Werner-Heisenberg-Institut),*
*Boltzmannstrasse 8, 85748 Garching bei München, Germany*

[4]*Arnold Sommerfeld Center for Theoretical Physics,*
*Ludwig-Maximilians-Universität München, 80333 München, Germany*

*E-mail:* acastellano@uchicago.edu, luest@mpp.mpg.de,
montella@mpp.mpg.de, zatti@mpp.mpg.de

ABSTRACT: We compute the supersymmetric entropy of the most general BPS black hole in 4d $\mathcal{N} = 2$ supergravity coupled to $n_V$ vector multiplets obtained from Type IIA string theory compactified on a Calabi–Yau threefold at large volume, including the all-genera leading-order $\alpha'$-corrections. These can be equivalently seen as D0-brane quantum effects from a dual five-dimensional M-theory perspective. We find that these corrections generically lead to both perturbative and non-perturbative contributions to the black hole entropy. We argue that the exception occurs for certain specific configurations where the gauge background, seen through the lens of D0-brane probes, behaves as purely electric or purely magnetic, thereby accounting for the absence of such non-perturbative effects. To explore this further, we perform a semiclassical analysis of the (non-)BPS particle dynamics in the near-horizon geometry of the underlying black hole, which is described by a maximally supersymmetric $\text{AdS}_2 \times \mathbf{S}^2$ solution. As a byproduct, this study provides additional insights into the non-perturbative stability of supersymmetric black hole geometries and suggests an interpretation in terms of complex saddles contributing to the worldline path integral.

# 1  Introduction

String theory provides a compelling framework for a consistent theory of quantum gravity. However, a complete understanding of its non-perturbative sector has remained so far elusive. In practice, our access to quantum gravitational physics often relies on a network of effective field theories valid in specific regimes and connected by dualities, which are believed to emerge from a common—but only partially understood—non-perturbative ultraviolet completion [1].

While the existence of such a UV-complete description can be effectively encoded within the language of effective field theory (EFT) through higher-derivative and higher-curvature corrections [2, 3] (see [4–8] for a modern perspective), genuinely non-perturbative phenomena are typically much more difficult to access. Nonetheless, there are physical settings where these effects become apparent, due to a pronounced and perhaps surprising mixing between ultraviolet and infrared degrees of freedom. A striking example is provided by the seminal work of Strominger and Vafa [9], who showed that the microscopic degeneracy of certain configurations of non-perturbative stringy BPS states precisely reproduces the leading-order entropy of an appropriate dual supersymmetric black hole system. This breakthrough was soon complemented by the works [10, 11], which extended the correspondence to also include one-loop corrections in the context of Type II/M-theory Calabi–Yau compactifications.

Yet, this is not the only way in which black holes may encode non-perturbative physics. Indeed, to go beyond the leading-order picture, one must incorporate quantum corrections. From the gravitational EFT perspective, perturbative contributions—organized as an infinite tower of higher-derivative terms—can be systematically accounted for via Wald's entropy formula [12, 13]. However, as emphasized in [14–17], even summing over all these perturbative effects may fail to fully reproduce the exact microscopic degeneracies. This mismatch has led to conjecture that truly non-perturbative corrections are essential for recovering a properly quantized black hole entropy from a macroscopic viewpoint. Importantly, such contributions can originate from several distinct physical mechanisms. For instance, [15] investigated corrections arising from (AdS) fragmentation processes and other euclidean saddle points [18–20]. In this work, we focus instead on a different class, namely non-perturbative effects associated to the presence of infinitely many (light) BPS particles in the UV theory. In our setting, these are realized as D0-branes propagating in the black hole background, whose quantum fluctuations give rise to exponentially suppressed contributions to the macroscopic entropy.

To investigate this question further, we consider a setup where we retain both computational control while exhibiting non-trivial and interesting dynamics. We thus focus on BPS black holes in four-dimensional $\mathcal{N} = 2$ supersymmetric effective field theories that arise from Calabi–Yau compactifications of Type IIA string theory. In this framework, the extremal black holes carry D0-D2-D4-D6 charges, depending on the nature of the non-perturbative objects sourcing them. Additionally, it has been argued [14, 21] that all relevant perturbative corrections to their quantum entropy formula are captured by the low-energy supergravity EFT through an infinite series of higher-derivative BPS operators involving the (anti-)self-dual components of the graviton and graviphoton field strengths, which are fully accessible.

The perturbative quantum corrections to the entropy then take the form of an asymptotic series entirely determined by the black hole charges [22–25]. In a previous work [26], it was demonstrated that this series can be exactly resumed within the large volume regime, thereby uncovering non-perturbative contributions. This approximation effectively selects corrections tied to constant momentum maps [27], which can be understood as arising from integrating out D0-branes [28, 29]. Although it is a well-known fact that asymptotic series may encode non-perturbative information [30–34], the physical interpretation of such contributions is not always straightforward. In particular, in the configurations studied in [26]—namely, D0-D2-D4 and D2-D6 systems—the precise resummation prescription revealed certain non-perturbative terms which nevertheless did not contribute eventually to the black hole entropy. Therefore, even though the resummed expressions non-trivially reproduce known limits beyond the original regime of validity of the perturbative series, the nature and role of the non-perturbative corrections remained somewhat unclear (see also [35, 36] for a recent discussion in the context of topological string theory).

In this work, we take a step forward by explicitly calculating all contributions to the entropy due to the D0-branes in the most general Calabi–Yau black hole BPS configuration. In addition, we also provide a direct connection between the presence of such non-perturbative terms and the dynamics of light charged particles in the near-horizon region of extremal 4d black holes. Through this, we deepen our understanding of the quantum effects in the aforementioned background, shedding some light on how these corrections impact the entropy of the black hole solutions in four-dimensional $\mathcal{N} = 2$ supergravity EFTs arising from string theory. Indeed, from an effective field theory standpoint, where some of the UV degrees of freedom may be even treated as particles [37], non-perturbative phenomena associated to them are expected to arise from their quantum backreaction—typically understood as a Schwinger pair production process. To test whether this interpretation applies in our context, two ingredients are thus essential. First, we need to identify black hole configurations where the entropy is genuinely sensitive to the non-perturbative corrections under consideration. Second, we must examine whether the dynamics of D0-brane probes in the near-horizon geometry can account for the presence or absence of these effects.

To address the first point, we analyze generic quantum-corrected systems carrying D0, D2, D4, and D6-brane charges. The resulting entropy we obtain for such configurations generalizes previous setups considered in the literature [21, 38, 39] and represents the first novel result of this work. We find that the only exceptions are certain special black holes with either D0-D2-D4 or D2-D6 charges. These are the only inequivalent configurations where no non-perturbative corrections due to D0-branes are present.

For the second issue, we conduct a semiclassical analysis, studying the motion of a generic charged particle in the near-horizon geometry of an extremal BPS black hole, specifically in a $\mathrm{AdS}_2 \times \mathbf{S}^2$ background with constant electric and magnetic fields [40, 41]. Our findings reveal that the on-shell trajectories followed by the particle depend on its effective charge-to-mass ratio. Notably, the relevant charge is determined by the relative phase between

the central charges of the probe particle and that of the black hole [42–44]. This picture reproduces the behavior of the non-perturbative contributions we compute. Specifically, we argue that the D0-brane corrections vanish for the D0-D2-D4 configuration because the phases of the particles and the background are aligned, such that the effective charge-to-mass ratio is saturated, leading to a cancellation of forces akin to the situation in flat Minkowski space. This implies the absence of backreaction at the level of the black hole metric. In all other cases, the BPS D0-branes become effectively subextremal in the $AdS_2$ throat, and the semiclassical trajectories followed by the particles cause them to fall inside the black hole. This conclusion follows from a quadratic constraint involving the resulting electric and magnetic couplings as well as their mass (in AdS units). Such particles are kinematically unable to discharge the black hole, hence suggesting that non-perturbative corrections due to BPS particles should not be viewed as a consequence of Schwinger pair production.

Instead, since the quantum-produced particles are not able to escape the near-horizon region, the non-perturbative effects should be rather understood as a renormalization due to vacuum polarization—i.e., the production of virtual states. Indeed, these contributions can be interpreted as real corrections to the Lagrangian encoding virtual processes that are inaccessible in the classical regime. This perspective also nicely explains why in the case of the D2-D6 configuration no such corrections are present, which follows from the fact that the background is seen as purely magnetic by the D0-branes, preventing any non-perturbative effect to occur. This proposal constitutes the main result of our work.

The paper is organized as follows. In Section 2, we present a structured and concise review of 4d $\mathcal{N} = 2$ BPS black holes in the presence of higher-derivative corrections. We focus on the low-energy EFT arising from Calabi–Yau compactifications of Type IIA string theory and discuss how the relevant perturbative quantum corrections are systematically encoded into the supergravity action. Subsequently, in Section 3, we compute the entropy of the most general quantum-corrected black hole configuration, namely a D0-D2-D4-D6 system—in the large volume regime. We also show how, starting from the general solution, one can employ certain symplectic transformations so as to reduce the full charge configuration to an equivalent D0-D2-D6 solution. The resulting black hole system generalizes previous results and serves as a useful background for investigating non-perturbative D0-brane effects. In Section 4, we turn to the question of whether non-perturbative corrections induced by the latter modify the quantum entropy formula. To do so, we first extract these corrections from a resummed topological string free energy [26] and show that they are generically present for arbitrary charge configurations, with notable exceptions such as D0-D2-D4 and D2-D6 black holes. To explain this behavior, we perform a detailed analysis of BPS probe particle dynamics in the near-horizon $AdS_2 \times \mathbf{S}^2$ geometry from the semiclassical perspective, uncovering a physical mechanism based on the effective extremality condition of the particle with respect to the AdS throat. We conclude in Section 5 with some final remarks and possible future directions.

## 2  4d $\mathcal{N} = 2$ Black Holes and Higher-Derivative Corrections

In this section we summarize our conventions as well as the main features of the framework we consider throughout this work, namely 4d $\mathcal{N} = 2$ setups arising from Type IIA string theory compactified on a Calabi–Yau threefold $X_3$ within the large volume approximation. We adopt the same conventions as in [26], to which we refer the reader interested in a more detailed and thorough exposition of these matters.

### 2.1  The two-derivative action

The bosonic part of the two-derivative action can be written as follows [45]

$$
S = \frac{1}{2\kappa_4^2} \int \mathcal{R} \star 1 + \frac{1}{2} \operatorname{Re} \mathcal{N}_{AB} F^A \wedge F^B + \frac{1}{2} \operatorname{Im} \mathcal{N}_{AB} F^A \wedge \star F^B
$$
$$
- \frac{1}{\kappa_4^2} \int G_{a\bar{b}} \, dz^a \wedge \star d\bar{z}^b + h_{pq} \, d\xi^p \wedge \star d\xi^q \,,
\tag{2.1}
$$

with $A, B = 0, 1, \ldots, h^{1,1}(X_3)$ and where we have denoted by $z^a = b^a + it^a$, $a = 1, \ldots, h^{1,1}(X_3)$, the scalar fields corresponding to the (complexified) Kähler deformations. The hypermultiplet moduli space, on the other hand, is parametrized by the $2 + 2h^{2,1}(X_3)$ complex coordinates $\xi^p$. We have chosen a basis of $U(1)$ gauge fields normalized so that they are integrally quantized, as customarily done in the literature. In the following, we restrict ourselves to the vector multiplet sector, since the black hole observables we will deal with depend only on those.

To simplify our task, let us choose a special set of projective coordinates $X^A$ [46–48] such that the Kähler moduli arise (locally) as simple quotients of the form $z^a = X^a/X^0$. Due to the constraints of $\mathcal{N} = 2$ supersymmetry, the entire geometry of the moduli space can be encoded into a holomorphic function $\mathcal{F}(X^A)$, usually referred to as the prepotential [48, 49]. This function is moreover homogeneous of degree two, meaning that it satisfies $\mathcal{F} = \frac{1}{2} X^A \mathcal{F}_A$, where $\mathcal{F}_A = \partial_{X^A} \mathcal{F}$. The vector moduli space is mathematically described as a projective special Kähler manifold [50–53], whose metric tensor $G_{a\bar{b}} = \partial_a \partial_{\bar{b}} K$ can be derived from the following Kähler potential

$$
K = -\log i \left( \bar{X}^A \mathcal{F}_A - X^A \bar{\mathcal{F}}_A \right) .
\tag{2.2}
$$

Similarly, the complexified gauge kinetic function $\mathcal{N}_{AB}$ appearing in (2.1) is completely determined by the Kähler structure moduli through the expression [47]

$$
\mathcal{N}_{AB} = \overline{\mathcal{F}}_{AB} + 2i \frac{(\operatorname{Im}\mathcal{F})_{AC} X^C (\operatorname{Im}\mathcal{F})_{BD} X^D}{X^C (\operatorname{Im}\mathcal{F})_{CD} X^D} \,,
\tag{2.3}
$$

where $\mathcal{F}_{KL} = \partial_{X^K} \partial_{X^L} \mathcal{F}$.

## 2.2 BPS higher-derivative corrections

Beyond the two-derivative level, these type of theories are known to present a rich structure of higher-dimensional and higher-curvature corrections. Some of these terms are furthermore $\frac{1}{2}$-BPS, which means that they are oftentimes protected from receiving certain quantum corrections due to supersymmetry (or more precisely holomorphy). This moreover implies that their exact moduli dependence can sometimes be determined. Using standard $\mathcal{N} = 2$ superspace notation, they can be written as integrals over half-superspace [27, 54–56][1]

$$\mathcal{L}_{\text{h.d.}} \supset -\frac{i}{2} \int d^4\theta \sum_{g \geq 1} \mathcal{F}_g(\mathcal{X}^A) \left(\mathcal{W}^{ij}\mathcal{W}_{ij}\right)^g + \text{h.c.}, \tag{2.4}$$

where $\mathcal{F}_g$ is an holomorphic function related to the $g$-loop topological free energy of the closed superstring, $\theta_\alpha$ denote the fermionic superspace coordinates (of negative chirality) and

$$\mathcal{W}_{\mu\nu}^{ij} = W_{\mu\nu}^{ij,-} - \mathcal{R}_{\mu\nu\rho\sigma}^{-}\theta^i\sigma^{\rho\sigma}\theta^j + \dots, \tag{2.5a}$$

$$\mathcal{X}^A = X^A + \frac{1}{2}\epsilon_{ij}\theta^i\sigma^{\mu\nu}\theta^j \left(F_{\mu\nu}^{A,-} - ie^{K/2}\bar{X}^A W_{\mu\nu}^{-}\right) + \dots, \tag{2.5b}$$

are the Weyl superfield [57, 58] and the reduced chiral superfields [59], respectively. The former transforms under the $SO(2)$ antisymmetric representation in the $i, j = 1, 2$, indices and moreover depends on the anti-self-dual components of the graviphoton field-strength [47]

$$W_{\mu\nu}^{-} = 2ie^{K/2}\text{Im}\,\mathcal{N}_{AB}X^A F_{\mu\nu}^{B,-}, \qquad W_{\mu\nu}^{ij,-} = \frac{\epsilon^{ij}}{2}W_{\mu\nu}^{-}, \tag{2.6}$$

as well as that of the Riemann tensor. Performing the integration over the fermionic variables, one obtains several terms entering in the bosonic action. For instance, upon combining the lowest (i.e., $\theta$-independent) components in the superfield expansion of $\mathcal{F}_g(\mathcal{X}_A)$ and $\mathcal{W}^{2g-2}$ with the $\theta^2$-term in (2.5a) squared, one obtains operators within (2.4) whose structure is [22, 58]

$$\mathcal{L}_{\text{h.d.}} \supset -\frac{i}{2}\sum_{g \geq 1} \mathcal{F}_g(X^A)\,\mathcal{R}_{-}^2\,W_{-}^{2g-2} + \text{h.c.}, \tag{2.7}$$

and where the precise index contractions appearing in (2.7) can be deduced from eqs. (2.4) and (2.5a). Let us also remark that not all the purely bosonic terms that can be extracted from the superspace Lagrangian (2.4) are quadratic in the Riemann tensor (see e.g., [22, 58]).

Interestingly, as explained in [28, 29], one can compute all (non-)perturbative stringy $\alpha'$-corrections in $\mathcal{F}_g(X^A)$ for $g \geq 0$ using the duality between Type IIA string theory on $X_3$ and M-theory compactified on the same threefold times a circle. This rests on the fact that the string coupling belongs to a hypermultiplet, which decouples from the vector multiplets at two-derivatives [60] and thus can be freely tuned. Hence, for a single hypermultiplet with

---

[1]The analogous $g = 0$ contribution gives the prepotential term in $\mathcal{N} = 2$ supergravity upon identifying $\mathcal{F}_0(\mathcal{X}^A) \equiv \mathcal{F}(\mathcal{X}^A)$ as functions of the chiral superfields (2.5b).

charges $(q'_A, p^{B\prime})$ and mass $m = |Z_p|$ in 4d Planck units, where $Z_p = e^{K/2}\left(p^{A\prime}\mathcal{F}_A - q'_A X^A\right)$ denotes its central charge, one indeed obtains a generating function via a Schwinger-like integral of the form [61, 62][2]

$$\sum_{g \geq 0} \delta\mathcal{F}_g^{(\mathrm{hyp})}(X^A)\, W_-^{2g-2} = -\frac{1}{4}\int_{i0^+}^{i\infty}\frac{\mathrm{d}\tau}{\tau}\frac{1}{\sin^2\frac{\tau W_- - e^{K/2}\bar{Z}_p}{2}}e^{-\tau|Z_p|^2}\,, \tag{2.8}$$

with the integration being fixed along the positive imaginary axis by causality [63] (in the form of the $i\epsilon$ prescription). Note that the coupling of the particle to the graviphoton involves the anti-holomorphic central charge, thereby ensuring the holomorphy of the end result [37].

## 2.3 Generalized attractor mechanism

An interesting class of geometrical objects that one can construct within these theories are supersymmetric black holes. A comprehensive treatment of this type of solutions can be found e.g., in [21, 64]. They moreover exhibit certain universal features, such as the stabilization of the moduli fields—which couple to the electromagnetic background turned on by the black hole charges $(q_A, p^B)$—at the horizon locus, according to the so-called *attractor mechanism* [40, 65–67]. Importantly for us, this analysis can be extended beyond the two-derivative level [22–24, 68–70], also including the higher-curvature corrections discussed in the previous section. This is what we review next.

For convenience, we introduce some rescaled quantities as follows [22, 68] (we henceforth omit the anti-self-dual subindices in the relevant field strengths)

$$Y^A = CX^A\,, \qquad \Upsilon = C^2 W^2\,, \qquad \text{with} \qquad C^2 = e^{\mathcal{K}}\,\bar{\mathscr{Z}}^2\,, \tag{2.9}$$

where $\mathscr{Z}$ defines a generalized black hole central charge (cf. eq. (2.16)) whilst $\mathcal{K}$ determines the a symplectic invariant combination which has a functional form reminiscent of the Kähler potential (cf. eq. (2.2))

$$\mathscr{Z} = e^{\mathcal{K}}\left(p^A F_A(X, W^2) - q_A X^A\right)\,, \tag{2.10a}$$

$$e^{-\mathcal{K}} = i\bar{X}^A F_A(X, W^2) - iX^A \bar{F}_A(\bar{X}, \bar{W}^2)\,. \tag{2.10b}$$

and we defined $F_A(X, W^2) = \partial_{X^A}F(X, W^2)$. Here, $F(X, W^2)$ denotes a generalization of the holomorphic prepotential associated to the underlying 4d $\mathcal{N} = 2$ theory that includes the effects of higher-derivative terms, namely

$$F(X, W^2) = \sum_{g=0}^{\infty} F_g(X^A)W^{2g}\,, \tag{2.11}$$

---

[2]It is worth stressing that, from the string theory perspective, one includes higher-genus contributions such that (2.8) actually contains terms that arise at different perturbative orders in $g_s$. On the contrary, from the 5d point of view, we in principle consider only one-loop effects. However, supersymmetry guarantees that the computation is exact in the field theory loop expansion—barring non-perturbative effects, see Section 4 below.

In the following, we will find convenient to rescale the generalized prepotential (2.11), such that [14]

$$F(Y, \Upsilon) := C^2 F(X, W^2) = \sum_{g=0}^{\infty} F_g(Y^A) \Upsilon^g \,, \tag{2.12}$$

where the last equality exploits the homogeneity properties of $F_g$, which follow themselves from an analogous relation satisfied by $F(Y, \Upsilon)$

$$2F(\Upsilon, Y^A) = 2\Upsilon F_\Upsilon + Y^A F_A \,. \tag{2.13}$$

The coefficients $F_g(X^A)$ can be directly related to topological closed string amplitudes [71–74]

$$F_g(Y^A) = (-1)^g \, 2^{-6g} \mathcal{F}_g(Y^A) \,, \tag{2.14}$$

where the $\mathcal{F}_g$ are the same holomorphic functions appearing in (2.4). In terms of the rescaled prepotential, the quantity $\mathscr{Z}$ becomes

$$\mathscr{Z} = \bar{\mathscr{Z}}^{-1} \left( p^A F_A(Y, \Upsilon) - q_A Y^A \right) \,, \tag{2.15}$$

and physically controls the warp factor of the metric in the black hole background [22, 41], whose near-horizon line element reads (using isotropic coordinates)

$$ds^2 = -e^{2U(y)} dt^2 + e^{-2U(y)} \left( dy^2 + y^2 d\Omega_2^2 \right) \,, \qquad \text{with } e^{-2U(y)} = \frac{|\mathscr{Z}|^2 \kappa_4^2}{8\pi y^2} \,, \tag{2.16}$$

thus also incorporating the effect of the higher-derivative chiral terms captured by eq. (2.4). The attractor equations then determine the values for the moduli fields $Y^A$ when evaluated at the horizon, which are fixed by [69, 70]

$$\begin{aligned} ip^A &= Y^A - \bar{Y}^A \,, \\ iq_A &= F_A(Y, \Upsilon) - \bar{F}_A(\bar{Y}, \bar{\Upsilon}) \,, \end{aligned} \tag{2.17}$$

whereas $\Upsilon$ is set to $-64$.

## 2.4 Quantum entropy formula for BPS black holes

Finally, let us state the *quantum entropy formula* for BPS black holes with the near-horizon geometry given by (2.16), which may be expressed as follows [22]

$$\mathcal{S}_{\mathrm{BH}} = \pi \left[ |\mathscr{Z}|^2 + 4\mathrm{Im} \left( \Upsilon \partial_\Upsilon F(Y, \Upsilon) \right) \right] \,, \tag{2.18}$$

and is therefore entirely determined by the black hole charges via (2.17). The first term in (2.18) coincides with the value of the horizon area divided by $4G_N$, hence providing for the Bekenstein-Hawking contribution to the entropy, whilst the second piece captures deviations from the area law. Importantly, we note that both terms are sensitive to the higher-derivative operators shown in (2.4).

The entropy (2.18) has been computed using Wald's prescription [12, 13] within the restricted framework of conformal off-shell $\mathcal{N}=2$ supergravity coupled to $n_V+1$ vector multiplets [50–52, 57, 75], which reduces to the more familiar 4d $\mathcal{N}=2$ (Poincaré) supergravity only after partial gauge fixing.[3] This formalism can be used, in turn, to derive the attractor equations (2.17) as well as the near-horizon metric (2.16). This means, consequently, that (2.18) provides the macroscopic entropy associated to BPS black hole solutions in Calabi–Yau compactifications of Type IIA string theory, when restricting ourselves to the gravity and vector multiplet sectors.[4] Notice, however, that upon doing so we might be missing some contributions (see [26] and references therein).

In [14] a detailed analysis of the origin of (2.18) was performed. They showed that the entropy (2.18) is the Legendre dual of the free energy $\mathrm{Im}\,F$

$$\mathcal{S}_{\mathrm{BH}} = 4\pi \left[ \mathrm{Im}\,F - \frac{1}{2} q_A \mathrm{Re}\,Y^A \right], \tag{2.19}$$

where $q_A$ and $\mathrm{Re}\,Y^A$ are, respectively, the electric charges and potentials of the black hole. There it was suggested that, in the context of the full Type IIA string theory, the free energy appearing in (2.19) should correspond to a protected supersymmetric index rather than the partition function. This idea has been supported by matching the black hole free energy with an indexed observable defined within the CFT living on the branes sourcing the BPS black hole background. In particular, the alternating signs of the terms which add up to give the index should induce cancellations which remove from the Type IIA BPS states degeneracy the dependence on the hypermultiplet vacuum expectation values (vevs).

In what follows, we will not be concerned about whether (2.18) and (2.19) are truly computing an entropy or the related supersymmetric index in Type IIA theory, and we will just focus on its properties. With this subtlety in mind, we will refer to (2.18) simply as the *BPS black hole entropy*.

## 3   The D0-D2-D4-D6 Quantum Black Hole Solution

The discussion in Section 2 was presented in a model independent way, thereby expressing every physical quantity in terms of an undetermined (generalized) prepotential. The latter is constrained to be both holomorphic and homogeneous of order two in the supergravity fields $\{X^A, W\}$, but is otherwise left arbitrary. In this section, we would like to make contact with string theory and particularize our discussion to the Type IIA large volume/radius regime, where simple and explicit formulae arise which are valid regardless of the specific Calabi–Yau

---

[3]The relation between conformal and Poincaré (extended) supergravity requires the introduction of an additional vector multiplet that can be used to gauge-fix dilatation invariance [50]. See also [76, 77] for early reviews on the topic.

[4]As is well known, the two-derivative theory can be consistently truncated. However, the quantum corrections to the hypermultiplet sector are not fully understood [55, 78], which prevents us from determining whether higher-derivative terms involving these fields would obstruct the truncation.

threefold that is chosen. As we review in Section 3.1, this is equivalent to considering the following generalized prepotential

$$F(Y, \Upsilon) = \frac{D_{abc} Y^a Y^b Y^c}{Y^0} + d_a \frac{Y^a}{Y^0} \Upsilon + G(Y^0, \Upsilon) , \qquad (3.1)$$

where $D_{abc}$ and $d_a$ are related to topological data of the underlying Calabi–Yau manifold and $G(Y^0, \Upsilon)$ encodes the higher-genus effects computed by (2.8). The main finding of this section is the determination of the entropy function (2.18) restricted to a prepotential of the form (3.1) in the case of a generic BPS black hole with arbitrary D0-D2-D4-D6 charges:

$$
\begin{aligned}
\mathcal{S}_{\mathrm{BH}} = {} & \frac{\pi}{3p^0} \sqrt{ \frac{4}{3} \left( \tilde{\Delta}_a \tilde{x}^a \right)^2 - 9 \left( p^0 p^A \tilde{q}_A - 2 D_{abc} p^a p^b p^c \right)^2 } \\
& + 4\pi \left[ \mathrm{Im}\, G - \mathrm{Re}\, Y^0 \mathrm{Im}\, G_0 - \frac{1}{\sqrt{3}} \frac{(\mathrm{Re}\, Y^0)^2}{|Y^0|^3} \tilde{x}^a d_a \Upsilon \right] .
\end{aligned}
\qquad (3.2)
$$

Here $\Upsilon = -64$ at the attractor point, and $\tilde{\Delta}$, $\tilde{x}^a$, $\tilde{q}_A$ and $G_0$ are all quantities that will be defined (sometimes implicitly) later on in terms of the gauge charges $(q_A, p^B)$ as well as the higher-derivative corrections to $F(Y, \Upsilon)$. This result may be regarded as a natural extension of older works [21, 38, 39], where the general D0-D2-D4-D6 black hole entropy was either obtained by restricting to the leading-order (classical) cubic prepotential, or rather by taking into account the above perturbative quantum corrections but without providing explicit formulae for the most general charge configuration.

We will also take the opportunity in this section to review how certain symplectic transformations can be used as solution generating techniques, allowing us to reduce the general D0-D2-D4-D6 black hole system to an equivalent setup carrying only D0-D2-D6 gauge charges. This will prove to be very useful later on in Section 4, when studying the presence or absence of non-perturbative D0-brane effects in the BPS quantum black hole entropy (3.2).

## 3.1 The large volume patch

The large radius approximation of Type IIA string theory compactified on a Calabi–Yau threefold $X_3$ is defined as the limit $z^a \to i\infty$ for all $a = 1, \ldots, h^{1,1}(X_3)$. In what follows, we show how the terms within the generalized prepotential (2.11) get simplified when evaluated at large volume. We will organize them according to their order within the genus-$g$ expansion.

For the genus-0 contribution, one obtains (using string units)

$$
\begin{aligned}
\mathcal{F}_0(X^A) = {} & -\frac{1}{6} \mathcal{K}_{abc} \frac{X^a X^b X^c}{X^0} + K_{ab}^{(1)} X^a X^b + K_a^{(2)} X^0 X^a + K^{(3)} (X^0)^2 \\
& - \frac{(X^0)^2}{(2\pi i)^3} \sum_{\boldsymbol{k} > 0} n_{\boldsymbol{k}}^{(0)} \, \mathrm{Li}_3 \left( e^{2\pi i k_a z^a} \right) .
\end{aligned}
\qquad (3.3)
$$

The cubic term corresponds to the well-known tree-level contribution and $\mathcal{K}_{abc}$ denote the triple intersection numbers of the threefold. The other quantities appearing above are also

determined by topological information of the compactification manifold. Indeed, introducing an integral basis of harmonic 2-forms $\{\omega_a\} \in H^{1,1}(X_3, \mathbb{Z})$, we have the explicit expressions

$$\mathcal{K}_{abc} = \omega_a \cdot \omega_b \cdot \omega_c \,, \qquad K_a^{(2)} = \frac{1}{24} c_2(TX_3) \cdot \omega_a \,, \qquad K^{(3)} = \frac{i\zeta(3)}{2(2\pi)^3} \chi_E(X_3) \,, \qquad (3.4)$$

where $\chi_E(X_3) = 2\, h^{1,1}(X_3) - 2\, h^{2,1}(X_3)$ corresponds the Euler characteristic of the threefold whilst $c_2(TX_3)$ denotes the second Chern class of its tangent bundle. The quantity $K_{ab}^{(1)}$ can be determined (up to monodromy) by requiring good symplectic transformation properties of the underlying period vector [79–82]. This in particular means that we can always set it to zero patch-wise by a proper choice of symplectic frame. Lastly, the integral coefficients $n_{\boldsymbol{k}}^{(0)}$ are known as genus-zero Gopakumar–Vafa invariants and count, for each positive homology representative $\boldsymbol{k} = k_a \gamma^a \in H_2^+(X_3, \mathbb{Z})$, the indexed degeneracy of supersymmetric D2-brane states wrapped on 2-cycles within the corresponding holomorphic 2-cycle class [28, 29].

The genus-1 topological string amplitude, on the other hand, can be expanded around the large radius point as follows [27, 54, 83]

$$\mathcal{F}_1(X^A) = \frac{1}{24} \int_{X_3} J_{\mathbb{C}} \wedge c_2(TX_3) + \mathcal{O}\left(e^{2\pi i z^a}\right) = \frac{1}{24} c_{2,a}\, z^a + \mathcal{O}\left(e^{2\pi i z^a}\right)\,, \qquad (3.5)$$

where $J_{\mathbb{C}} = z^a\, \omega_a = (b^a + it^a)\, \omega_a$ is the complexified Kähler 2-form of the Calabi–Yau threefold. This contribution may be easily understood as coming from the dimensional reduction of the analogous $t_8 t_8 \mathcal{R}^4$ operator in 10d $\mathcal{N} = (1,1)$ supergravity [84, 85].

For higher-genus terms, the leading contribution corresponds to constant maps from the worldsheet to the Calabi–Yau threefold [27]. These can be equivalently determined from the dual M-theory perspective via a Schwinger one-loop calculation associated to the tower of D0 bound states, whose masses are given in flat space by $m_n = 2\pi |n|\, m_s/g_s$, where $n \in \mathbb{Z}$ is the D0-brane charge. Therefore, substituting this into (2.8) and performing the integral as well as the infinite sum, one finds

$$\mathcal{F}_{g>1}(X^A) \supset \frac{\chi(X_3)}{2}(-1)^{g-1} 2(2g-1) \frac{\zeta(2g)\zeta(3-2g)}{(2\pi)^{2g}}\, (X^0)^{2-2g}\,, \qquad (3.6)$$

where we have taken into account that each D0-brane yields $-\frac{\chi(X_3)}{2}$ times the contribution of a single hypermultiplet [29]. Notice that this precisely matches the dominant result along this limit [28, 86].

Putting everything together, we thus conclude that the generalized prepotential (2.12), when expanded around the large volume point, can be well-approximated by the function

$$F(Y, \Upsilon) = \frac{D_{abc} Y^a Y^b Y^c}{Y^0} + d_a \frac{Y^a}{Y^0} \Upsilon + G(Y^0, \Upsilon) + \mathcal{O}\left(e^{2\pi i z^a}\right)\,, \qquad (3.7)$$

where $D_{abc} = -\frac{1}{6}\mathcal{K}_{abc}$ and $d_a = -\frac{1}{24}\frac{1}{64}c_{2,a}$ are related to topological data of the underlying Calabi–Yau threefold and the function $G(Y^0, \Upsilon)$ corresponds to the one-loop determinant

(2.8) of the D0-branes. Notice that linear and quadratic terms in $X^a$ have been removed by a proper choice of symplectic frame.[5] In turn, $G(Y^0, \Upsilon)$ may be written more compactly as

$$G(Y^0, \Upsilon) = -\frac{i}{2(2\pi)^3} \chi_E(X_3) (Y^0)^2 \sum_{g=0,2,3,\dots} c^3_{g-1} \alpha^{2g} + \dots, \tag{3.8}$$

where we defined[6]

$$c^3_{g-1} = (-1)^{g-1} 2(2g-1) \frac{\zeta(2g)\zeta(3-2g)}{(2\pi)^{2g}}, \qquad \alpha^2 = -\frac{1}{64} \frac{\Upsilon}{(Y^0)^2}. \tag{3.9}$$

The ellipsis in (3.8), on the other hand, are meant to indicate that there would be a priori further non-analytic corrections around $\alpha = 0$ [26]. Notice that the first two terms in (3.7) capture the leading-order contribution to $F(Y, \Upsilon)$ at $g = 0, 1$, respectively.

## 3.2 The explicit derivation

In this section, we want to follow the same steps as in [38] in order to evaluate the entropy for the most general BPS black hole solution existing in the Type IIA theory, taking into account as well the quantum corrections studied in [26] (see also references therein). We henceforth assume the prepotential to have the explicit form (3.1).

Let us start by recalling the attractor relations (2.17)

$$q_a = 2 \operatorname{Im} F_a, \tag{3.10a}$$

$$q_0 = 2 \operatorname{Im} F_0, \tag{3.10b}$$

$$p^a = 2 \operatorname{Im} Y^a, \tag{3.10c}$$

$$p^0 = 2 \operatorname{Im} Y^0, \tag{3.10d}$$

and consider first the set of equations (3.10a). Isolating the $\operatorname{Re} Y^a$ terms, the latter can be written as follows

$$D_{abc} \left( \operatorname{Re} Y^b - \frac{\operatorname{Re} Y^0}{p^0} p^b \right) \left( \operatorname{Re} Y^c - \frac{\operatorname{Re} Y^0}{p^0} p^c \right) = \Delta_a, \tag{3.11}$$

with

$$\Delta_a = |Y^0|^2 \left( -\frac{\tilde{q}_a}{3p^0} + \frac{1}{(p^0)^2} D_{abc} p^b p^c \right), \qquad \tilde{q}_a \equiv q_a + p^0 \frac{d_a \Upsilon}{|Y^0|^2}. \tag{3.12}$$

Defining $x^a$ as the solution (if it exists) of the algebraic system $D_{abc} x^b x^c = \Delta_a$, we obtain for each rescaled modulus $Y^a$ the following expression

$$\operatorname{Re} Y^a = x^a + \frac{\operatorname{Re} Y^0}{p^0} p^a. \tag{3.13}$$

---

[5]See Section 3.3 for more details on symplectic transformations.

[6]The notation is chosen to reflect the fact that $c^3_{g-1}$ actually corresponds to the (integrated) third power of the Chern class associated to the Hodge bundle over the moduli space of Riemann surfaces of genus $g$ [27, 87].

Consider now eq. (3.10b). After moving to the left-hand side all the higher-genus contributions (including the genus-1 term), one finds

$$\tilde{q}_0 = 2\text{Im}\left[-\frac{1}{(Y^0)^2}D_{abc}Y^aY^bY^c\right],\tag{3.14}$$

with

$$\tilde{q}_0 \equiv q_0 + 2\text{Im}\left[\frac{1}{(Y^0)^2}d_aY^a\Upsilon\right] - 2\text{Im}\,G_0\,.\tag{3.15}$$

Upon massaging a bit the expression shown in (3.14), we eventually arrive at

$$\left(p^0p^A\tilde{q}_A - 2D_{abc}p^ap^bp^c\right)|Y^0|^4 = 2(p^0)^3\text{Re}\,Y^0x^a\Delta_a\,.\tag{3.16}$$

For future reference, we summarize here the algebraic steps one needs to follow so as to obtain the precise form of (3.16). First, one expands carefully the imaginary part of the right-hand side of (3.14). Then we replace the definition of $\text{Re}\,Y^a$ given by eq. (3.13), and we group terms together depending on the number of $x^a$ factors they include. Accordingly, terms cubic in $x^a$ are replaced by $x^a\Delta_a$ upon using the algebraic attractor equations. The quadratic ones are simplified by using the defining relation of $x^a$ (i.e., $D_{abc}x^bx^c = \Delta_a$) and subsequently replacing the explicit form of $\Delta_a$ (cf. (3.12)). The result is grouped together with other contributions with no dependence on $x^a$. Finally, the terms linear in $x^a$ are seen to cancel.

Let us now evaluate the BPS black hole entropy (2.19). We get

$$\begin{aligned}\frac{\mathcal{S}_{\text{BH}}}{4\pi} = &-2\text{Im}\left(\frac{1}{Y^0}\right)\Delta_a\text{Re}\,Y^a + \text{Im}\,G - \frac{1}{2}q_0\text{Re}\,Y^0\\&+\frac{1}{2}\text{Re}\left(\frac{1}{Y^0}\right)p^a\left(\frac{q_a}{6\text{Im}\left(\frac{1}{Y^0}\right)} + \frac{2}{3}d_a\Upsilon\right).\end{aligned}\tag{3.17}$$

In order to obtain (3.17), we first consider (2.19), then expand $\text{Im}\,F$ and subsequently replace (3.13). We write the terms cubic in $x^a$ again as $x^a\Delta_a$. Those which are quadratic in $x^a$ cancel out, whereas the linear terms survive. However, the latter multiply a quantity that is proportional to $\Delta_a$ and we end up with an expression which is made by a term proportional to $\Delta_ax^a$ as well as another one with no explicit dependence on $x^a$. The last manipulation we perform consists in absorbing some of the terms with no $x^a$ into the combination $\Delta_a\text{Re}\,Y^a$.

However, to make contact with the classical (i.e., two-derivative) solution [38], one would like to write the entropy more compactly in terms of a square root. Hence, we write (3.17) as

$$\frac{\mathcal{S}_{\text{BH}}}{4\pi} = \frac{1}{8\text{Re}\,Y^0}\left(p^0p^A\tilde{q}_A - 2D_{abc}p^ap^bp^c\right) + \text{Im}\,G + \frac{\text{Re}\,Y^0}{2p^0}p^A\delta q_A\,,\tag{3.18}$$

where we introduced $\delta q_A = \tilde{q}_A - q_A$. To obtain (3.18) from (3.17) we start by isolating the $x^a$-contribution inside $\text{Re}\,Y^a$. The term containing $\Delta_a$ but no $x^a$ is replaced by inserting the

definition of $\Delta_a$, and we used (3.16) to get rid of the quantity $\Delta_a x^a$. To proceed further, we need to use (3.16) so as to extract $|Y^0|^2$. We thus introduce the rescaled quantities

$$x^a = \tilde{x}^a \frac{|Y^0|}{\sqrt{3}p^0}\,, \qquad \tilde{\Delta}_a = D_{abc}\tilde{x}^b\tilde{x}^c = \Delta_a \frac{3(p^0)^2}{|Y^0|^2}\,, \tag{3.19}$$

such that in terms of these, (3.16) reads as follows

$$\left(p^0 p^A \tilde{q}_A - 2D_{abc}p^a p^b p^c\right)|Y^0| = \frac{2}{3\sqrt{3}}\text{Re}\,Y^0 \tilde{x}^a \tilde{\Delta}_a\,. \tag{3.20}$$

We now solve for $(\text{Re}\,Y^0)^2$ ignoring the implicit dependence on $|Y^0|$.[7] Adding $(p^0)^2/4$ to the solution for $(\text{Re}\,Y^0)^2$, we find

$$|Y^0|^2 = \frac{(p^0)^2}{3} \frac{\left(\tilde{\Delta}_a\tilde{x}^a\right)^2}{\frac{4}{3}\left(\tilde{\Delta}_a\tilde{x}^a\right)^2 - 9\left(p^0 p^A \tilde{q}_A - 2D_{abc}p^a p^b p^c\right)^2}\,. \tag{3.21}$$

Let us note that this expression only solves for $|Y^0|$ implicitly. In fact, the right-hand side also depends on $Y^0$—even if its leading order piece, namely the two-derivative contribution, does not. Still, follwing the same logic as in [26], one may use (3.21) to determine recursively (i.e., order by order in the perturbative expansion) the solution for $Y^0$.

Finally, we can use eq. (3.20) to remove $\text{Re}\,Y^0$ from the entropy (3.18) and subsequently insert the square root of (3.21) to simplify the residual $|Y^0|$ dependence. Thus, replacing the explicit expression of $\delta q_A$ defined around eq. (3.18), we end up with[8]

$$\mathcal{S}_{\text{BH}} = \frac{\pi}{3p^0}\sqrt{\frac{4}{3}\left(\tilde{\Delta}_a\tilde{x}^a\right)^2 - 9\left(p^0 p^A \tilde{q}_A - 2D_{abc}p^a p^b p^c\right)^2} \\ + 4\pi\left[\text{Im}\,G - \text{Re}\,Y^0\text{Im}\,G_0 - \frac{1}{\sqrt{3}}\frac{(\text{Re}\,Y^0)^2}{|Y^0|^3}\tilde{x}^a d_a \Upsilon\right]\,. \tag{3.22}$$

### 3.2.1 Comparison with previous results

In what follows, we would like to test our final entropy formula (3.22) by studying its compatibility with previous results existing in the literature.

---

[7]The higher-genus corrections ($g \geq 1$) introduce some dependence on $\text{Re}\,Y^0$ within $\tilde{q}_A$, $\tilde{x}^a$, and $\tilde{\Delta}_a$.

[8]Notice that $\tilde{x}^a$ is defined up to a sign, and we can always choose conventions such that $\tilde{\Delta}_a\tilde{x}^a > 0$. In general, in front of the square root there should appear a sign $s$

$$s = \frac{\tilde{\Delta}_a\tilde{x}^a}{\sqrt{(\tilde{\Delta}_a\tilde{x}^a)^2}}\,,$$

to ensure that the entropy is positive definite and hence well-defined.

### The classical D0-D2-D4-D6 solution

We can easily extract the two-derivative entropy function of a black hole with generic D0-D2-D4-D6 charges by simply setting $d_a = G = 0$ in all relevant formulae from the previous section. Doing so, we get

$$\mathcal{S}_{\mathrm{BH}} = \frac{\pi}{3p^0} \sqrt{\frac{4}{3} \left( \tilde{\Delta}_a \tilde{x}^a \right)^2 - 9 \left( p^0 p^A q_A - 2 D_{abc} p^a p^b p^c \right)^2} \,, \tag{3.23}$$

in agreement with [38, eq. (12)], once we identify our $\tilde{\Delta}_a$ with $\Delta_I$ defined therein. Notice that their $x^a$- and $\tilde{x}^a$-variables precisely match with ours here.

### The D2-D6 system

This solution can be obtained by setting $p^a = q_0 = 0$. The entropy (3.17) then yields

$$\mathcal{S}_{\mathrm{BH}} = -\frac{4\pi}{3} Y^a \left( q_a + \frac{4 d_a \Upsilon}{p^0} \right) + 4\pi \mathrm{Im}\, G \,, \tag{3.24}$$

which precisely reproduces the result obtained in [26] once we use that $\Upsilon = -64$ as well as $d_a = -\frac{1}{24}\frac{1}{64} c_a$.

### The D0-D2-D4 system

Finally, we study the $p^0 \to 0$ limit, i.e., the black string system described in [26, Section 3]. Interestingly, it becomes actually possible to take such a limit despite the apparent divergences seen in the entropy (3.2), which end up canceling out exactly. To show this, we need to expand all the quantities appearing in the entropy in terms of $p^0$. In particular, we have to solve (3.11) order by order in the aforementioned expansion parameter. We get

$$\tilde{x}^a = -\sqrt{3} p^a + p^0 \left[ \frac{\sqrt{3}}{6} D^{ab} q_b \right] + (p^0)^2 \frac{\sqrt{3}}{2} \left[ \frac{p^a}{4(\mathrm{Re}\, Y^0)^2} - \frac{1}{(\mathrm{Re}\, Y^0)^3} \Gamma^a \right] + \dots \,, \tag{3.25}$$

$$\tilde{\Delta}_a = 3 D_{abc} p^b p^c - p^0 q_a - (p^0)^2 \frac{d_a \Upsilon}{(\mathrm{Re}\, Y^0)^2} + \dots \,, \tag{3.26}$$

$$\tilde{q}_a = q_a + p^0 \frac{d_a \Upsilon}{(\mathrm{Re}\, Y^0)^2} + \dots \,, \tag{3.27}$$

$$\tilde{q}_0 = q_0 - 2\mathrm{Im}\, G_0 + \frac{p^a d_a \Upsilon}{(\mathrm{Re}\, Y^0)^2} + \dots \,, \tag{3.28}$$

where the dots indicate terms which are higher order in $p^0$, $D^{ab}$ is the inverse of $D_{ab} \equiv D_{abc} p^c$ and the quantities $\Gamma^a$ satisfy the relation

$$D_{abc} p^a p^b \Gamma^c = \mathrm{Re}\, Y^0 \left[ -\frac{1}{36} (\mathrm{Re}\, Y^0)^2 + \frac{1}{4} D_{abc} p^a p^b p^c - \frac{1}{3} p^a d_a \Upsilon \right] + \mathcal{O}(p^0) \,. \tag{3.29}$$

Inserting these expansions into (3.2), taking the limit and replacing $\operatorname{Re} Y^0 = Y^0$, we obtain

$$\frac{\mathcal{S}_{\mathrm{BH}}}{4\pi} = \frac{1}{2}\sqrt{D_{abc}p^a p^b p^c \left[\hat{q}_0 - 2\operatorname{Im} G_0 + \frac{p^a d_a \Upsilon}{(Y^0)^2}\right]} + \operatorname{Im} G - Y^0 \operatorname{Im} G_0 + \frac{p^a d_a \Upsilon}{Y^0}\,. \tag{3.30}$$

where we introduced the notation $\hat{q}_0 = q_0 + \frac{1}{12}D^{ab}q_a q_b$. Note that the attractor equation for $q_0$ in the $p^0 \to 0$ case reduces to (cf. [26, eq. (3.11)])

$$(Y^0)^2 = \frac{\frac{1}{4}D_{abc}p^a p^b p^c - d_a p^a \Upsilon}{\hat{q}_0 + i(G_0 - \bar{G}_0)}\,, \tag{3.31}$$

such that (3.30) eventually simplifies to[9]

$$\begin{aligned}
\frac{\mathcal{S}_{\mathrm{BH}}}{4\pi} &= -\frac{1}{4}\frac{D_{abc}p^a p^b p^c}{Y^0} + \operatorname{Im} G - Y^0 \operatorname{Im} G_0 + \frac{p^a d_a \Upsilon}{Y^0} \\
&= -\hat{q}_0 Y^0 + \operatorname{Im} G + Y^0 \operatorname{Im} G_0\,,
\end{aligned} \tag{3.32}$$

in perfect agreement with [26, eq. (3.14b)].

### 3.2.2 A comment on the genus-1 corrections

Before proceeding, let us take the chance to briefly comment here on the nature of the quantum corrections we have been dealing with up to now, especially in comparison with older classic results. In particular, in [38] it was argued that, starting from the Bekenstein-Hawking expression (3.23), one can easily extract certain corrections in the entropy due to a topological term in the prepotential that depends on the second Chern class of the threefold in a similar way than the genus-1 contribution (3.5) does. There it was shown that all one has to do is perform a shift on the electric charges, analogous to what we did in eqs. (3.12) and (3.15). However, it turns out that the reason why implementing such corrections is so easy hinges on the fact that they are not truly quantum corrections, but rather can be encoded into a *classical* prepotential of the form

$$F(Y^A) = D_{abc}\frac{Y^a Y^b Y^c}{Y^0} + K_a^{(2)}Y^0 Y^a\,, \tag{3.33}$$

where $K_a^{(2)}$ was defined in (3.4). Notice that this quantity exactly equals the (leading-order) genus-1 coefficient, but it is nevertheless very different in nature. First of all, its graviphoton dependence signals that it belongs to the genus-0 amplitude. Secondly, the reason why its effect amounts to a simple shift in the D2-brane charges is because such contribution can be easily generated from the cubic piece in $F(Y^A)$ by means of some symplectic transformation (cf. Section 3.3). Crucially, the correction-terms in the prepotential considered in this work

$$F(Y^A, \Upsilon) = \frac{D_{abc}Y^a Y^b Y^c}{Y^0} + d_a \frac{Y^a}{Y^0}\Upsilon + G(Y^0, \Upsilon)\,, \tag{3.34}$$

---

[9]Recall that in our conventions $Y^0 > 0$ and $D_{abc}p^a p^b p^c < 0$ [26].

cannot be obtained from the tree-level cubic piece via any symplectic map. Consequently, our quantum entropy formula (3.2) cannot be directly retrieved from that already derived in [38], and it therefore captures physical effects beyond the latter classical result.

One can verify this claim by checking that, if we consider the entropy found in [38] and we perform the shifts (3.12) and (3.15), we do not obtain the correct central charge $|\mathscr{Z}|^2$, i.e., the area law piece of the entropy for the generalized prepotential (3.1). For instance, in the black string (D0-D2-D4 system), one finds the following (generalized) central charge [26]

$$|\mathscr{Z}|^2 = -\frac{D_{abc}p^a p^b p^c - 2d_a p^a \Upsilon}{Y^0} - 2Y^0 \mathrm{Im}\, G_0 \,. \tag{3.35}$$

Instead, the entropy generated by shifting (3.23) as in in eqs. (3.12) and (3.15) reproduces just the first term in (3.32), namely

$$\frac{\mathcal{S}_{\mathrm{BH}}^{\mathrm{shift}}}{\pi} = 2\sqrt{D_{abc}p^a p^b p^c \left[\hat{q}_0 - 2\mathrm{Im}\, G_0 + \frac{p^a d_a \Upsilon}{(Y^0)^2}\right]} = -\frac{D_{abc}p^a p^b p^c}{Y^0} \neq |\mathscr{Z}|^2 \,. \tag{3.36}$$

Hence, this procedure crucially misses various terms both within the quantum-corrected area-law as well as deformations thereof, since writing (3.32) as in (2.18) one finds

$$\frac{\mathcal{S}_{\mathrm{BH}}}{\pi} = \left[-\frac{D_{abc}p^a p^b p^c}{Y^0} + 2\frac{p^a d_a \Upsilon}{Y^0} - 2Y^0 \mathrm{Im}\, G_0\right] + \left[4\mathrm{Im}\, G - 2Y^0 \mathrm{Im}\, G_0 + 2\frac{p^a d_a \Upsilon}{Y^0}\right] \,. \tag{3.37}$$

## 3.3 Symplectic transformations as a solution generating technique

As is well-known, the duality group of 4d $\mathcal{N} = 2$ supergravity coupled to $n_V$ vector multiplets corresponds to $Sp(2n_V + 2, \mathbb{R})$ [51, 88]. These transformations, which generalize the electric-magnetic invariance of the familiar 4d free Maxwell theory, rotate the abelian field strengths and their magnetic duals in a way that exchanges the equations of motion with the Bianchi identities [89]. In particular, defining

$$F_{\mu\nu}^{A,-} = \frac{1}{2}\left(F_{\mu\nu}^A + i \star F_{\mu\nu}^A\right) \,, \qquad G_A^- = \bar{\mathcal{N}}_{AB} F^{B,-} \,, \tag{3.38}$$

for the anti-self dual components of the vectors and their duals, respectively [47], and using the fact that the kinetic term for the $U(1)$ bosons in (2.1) can be written as

$$S \supset \frac{1}{4\kappa_4^2} \int F^{A,-} \wedge G_A^- + \mathrm{h.c.} \,, \tag{3.39}$$

it becomes apparent that the equations of motion are left invariant under the transformation

$$\begin{pmatrix} F^{A,-} \\ G_A^- \end{pmatrix} \longrightarrow \begin{pmatrix} \mathcal{A} & \mathcal{B} \\ \mathcal{C} & \mathcal{D} \end{pmatrix} \begin{pmatrix} F^{A,-} \\ G_A^- \end{pmatrix} \,, \tag{3.40}$$

where the above constant $(n_V + 1) \times (n_V + 1)$ matrices must satisfy the constraint relations

$$\mathcal{A}^{\mathrm{T}} \mathcal{D} - \mathcal{C}^{\mathrm{T}} \mathcal{B} = \mathbb{1}_{n_V+1} \,, \qquad \mathcal{A}^{\mathrm{T}} \mathcal{C} - \mathcal{C}^{\mathrm{T}} \mathcal{A} = \mathcal{B}^{\mathrm{T}} \mathcal{D} - \mathcal{D}^{\mathrm{T}} \mathcal{B} = 0 \,. \tag{3.41}$$

From here, one derives the transformation law for the gauge kinetic matrix $\mathcal{N}_{AB}$

$$\mathcal{N}_{AB} \rightarrow \left(\mathcal{D}_A{}^C \mathcal{N}_{CD} + \mathcal{C}_{AD}\right) \left[(\mathcal{C} + \mathcal{B}\mathcal{N})^{-1}\right]^D{}_B , \tag{3.42}$$

which can be reproduced via the same symplectic rotation

$$\begin{pmatrix} X^A \\ \mathcal{F}_A \end{pmatrix} \longrightarrow \begin{pmatrix} X^{A\prime} \\ \mathcal{F}'_A \end{pmatrix} = \begin{pmatrix} \mathcal{A} & \mathcal{B} \\ \mathcal{C} & \mathcal{D} \end{pmatrix} \begin{pmatrix} X^A \\ \mathcal{F}_A \end{pmatrix} , \tag{3.43}$$

acting now on the period vector $\mathbf{\Pi} = (X^A, \mathcal{F}_A)^{\mathrm{T}}$. Furthermore, one can show that the transformed periods shall be obtained from a new prepotential function [47, 90]

$$\begin{aligned} F'(X^{A\prime}) &= F(X^A) - \frac{1}{2} X^A \mathcal{F}_A + \frac{1}{2} X^A (\mathcal{C}^{\mathrm{T}}\mathcal{B} + \mathcal{A}^{\mathrm{T}}\mathcal{D})_A{}^B \mathcal{F}_B \\ &+ \tfrac{1}{2} X^A (\mathcal{C}^{\mathrm{T}}\mathcal{A})_{AB} X^B + \tfrac{1}{2} \mathcal{F}_A (\mathcal{D}^{\mathrm{T}}\mathcal{B})^{AB} \mathcal{F}_B . \end{aligned} \tag{3.44}$$

which may be easily derived by direct integration of (3.43).

Interestingly, the above story can be generalized in the presence of a chiral background field [91, 92]. Recall that this is precisely the situation of interest in conformal supergravity, where one treats the square of the Weyl multiplet as one such background (see discussion around (2.6)). The new central object of the theory becomes the generalized prepotential, whose transformation law reads [91]

$$\begin{aligned} F'(Y^{A\prime}) &= F(Y^A) - \frac{1}{2} Y^A F_A + \frac{1}{2} Y^A (\mathcal{C}^{\mathrm{T}}\mathcal{B} + \mathcal{A}^{\mathrm{T}}\mathcal{D})_A^B F_B \\ &+ \tfrac{1}{2} Y^A (\mathcal{C}^{\mathrm{T}}\mathcal{A})_{AB} Y^B + \tfrac{1}{2} F_A (\mathcal{D}^{\mathrm{T}}\mathcal{B})^{AB} F_B , \end{aligned} \tag{3.45}$$

and is thus identical to (3.44) upon substituting the periods with their rescaled analogues, namely $\mathbf{\Pi} \rightarrow C\mathbf{\Pi} = (Y^A, F_A)^{\mathrm{T}}$ (cf. Section 2.3 for details). The important point for our purposes here is to note that the quantum black hole entropy (2.18) defines a symplectic function,[10] meaning that it is consistent with the electric-magnetic invariance of the theory and therefore does not depend on the particular symplectic frame we choose. This implies, in turn, that one may exploit the duality symmetries of the theory to relate the entropy function of certain black hole systems to other, equivalent ones related via the transformation (3.43).

For us, the relevant symplectic map will be the following (we assume $p^0 \neq 0$)

$$Y^{0\prime} = Y^0 , \qquad Y^{a\prime} = Y^a + \frac{\gamma^a}{p^0} Y^0 , \tag{3.46}$$

which is parametrized by the real quantities $\gamma^a$. This transformation, when expressed in terms of the special coordinates $z^a := X^a = Y^a/Y^0$, amounts to performing $z^a \rightarrow z^a + \gamma^a/p^0$,

---

[10]This crucially depends on the fact that $\Upsilon F_\Upsilon = F(\Upsilon, Y^A) - \frac{1}{2} Y^A F_A$ is also a symplectic function [22, 91].

and simply corresponds to the axionic shift symmetry that the theory is known to exhibit. The analogous law for the magnetic periods reads [93]

$$F_0' = F_0 - \frac{\gamma^a F_a}{p^0} - 3D_{abc}\frac{\gamma^a\gamma^b Y^c}{(p^0)^2} - D_{abc}\frac{\gamma^a\gamma^b\gamma^c Y^0}{(p^0)^3}\,,$$

$$F_a' = F_a + 6D_{abc}\frac{Y^b\gamma^c}{p^0} + 3D_{abc}\frac{\gamma^b\gamma^c Y^0}{(p^0)^2}\,. \tag{3.47}$$

Notice that the anti-self dual part of the graviphoton field strength is inert under symplectic transformations, since it belongs to another (i.e., the gravity) supermultiplet [47]. Therefore, one has $\Upsilon' = \Upsilon$ when applying the linear map specified by eqs. (3.46)-(3.47). At the same time, the attractor relations (cf. eqs. (3.10a)-(3.10d)) imply the following charge dictionary

$$p^{0\prime} = p^0\,, \qquad p^{a\prime} = p^a + \gamma^a\,, \qquad q_a' = q_a + 6D_{abc}\frac{p^b\gamma^c}{p^0} + 3D_{abc}\frac{\gamma^b\gamma^c}{p^0}\,,$$

$$q_0' = q_0 - \frac{\gamma^a q_a}{p^0} - 3D_{abc}\frac{\gamma^a\gamma^b p^c}{(p^0)^2} - D_{abc}\frac{\gamma^a\gamma^b\gamma^c}{(p^0)^2}\,. \tag{3.48}$$

Consequently, under this set of transformations, the quantities appearing in the black hole entropy (2.18) get modified according to[11]

$$p^A F_A \to p^A F_A + \Delta\,, \qquad q_A Y^A \to q_A Y^A + \Delta\,, \qquad \text{Im}(\Upsilon\partial_\Upsilon F) \to \text{Im}(\Upsilon\partial_\Upsilon F)\,, \tag{3.49}$$

with

$$\Delta = 6D_{abc}\frac{p^a Y^b\gamma^c}{p^0} + 3D_{abc}\frac{\gamma^a Y^b\gamma^c}{p^0} + 3D_{abc}\frac{p^a\gamma^b\gamma^c Y^0}{(p^0)^2} + 2D_{abc}\frac{\gamma^a\gamma^b\gamma^c Y^0}{(p^0)^2}\,, \tag{3.50}$$

so that we end up having the chain of equalities

$$\mathcal{S}_{\text{BH}}'[p^{0\prime}, p^{a\prime}, q_a', q_0'] = \mathcal{S}_{\text{BH}}[p^{0\prime}, p^{a\prime}, q_a', q_0'] = \mathcal{S}_{\text{BH}}[p^0, p^a, q_a, q_0]\,, \tag{3.51}$$

where the charges are related as in (3.48). Notice, in particular, the first equality in (3.51), which states that the transformed black hole entropy preserves the same functional form as the original one. This follows from the fact that the map (3.46)-(3.47) corresponds to an actual *symmetry* of the theory, as we demonstrate in the following. To show this, we first write the transformation as a square matrix $\mathsf{M}$ acting on the generalized period vector

$$\begin{pmatrix} Y^{0\prime} \\ Y^{a\prime} \\ F_0' \\ F_a' \end{pmatrix} = \begin{pmatrix} 1 & 0 & 0 & 0 \\ \frac{\gamma^a}{p^0} & \delta_b^a & 0 & 0 \\ -D_{abc}\frac{\gamma^a\gamma^b\gamma^c}{(p^0)^3} & -3D_{bcd}\frac{\gamma^c\gamma^d}{(p^0)^2} & 1 & -\frac{\gamma^a}{p^0} \\ 3D_{abc}\frac{\gamma^b\gamma^c}{(p^0)^2} & 6D_{abc}\frac{\gamma^c}{(p^0)^2} & 0 & \delta_a^b \end{pmatrix} \begin{pmatrix} Y^0 \\ Y^b \\ F_0 \\ F_b \end{pmatrix}\,, \tag{3.52}$$

---

[11]Notice that the invariance of the piece capturing the deviation from the area law $\text{Im}(\Upsilon\partial_\Upsilon F)$ under the symplectic map (3.46) rests on the reality condition of the parameters $\gamma^a$.

such that it clearly verifies

$$\mathsf{M}^{\mathrm{T}}\eta\,\mathsf{M} = \eta \equiv \begin{pmatrix} 0 & \mathbb{1}_{h^{1,1}+1} \\ -\mathbb{1}_{h^{1,1}+1} & 0 \end{pmatrix}, \tag{3.53}$$

and thus belongs to the symplectic group $Sp(2h^{1,1}(X_3) + 2, \mathbb{R})$. As already stressed, this is precisely the reason why the black hole entropy $\mathcal{S}_{\mathrm{BH}}$ remains the same, since the former is defined as a sum of two quantities

$$|\mathscr{Z}|^2 = \mathbf{q}^{\mathrm{T}} \cdot \eta \cdot C\mathbf{\Pi}, \qquad \mathrm{Im}\left(\Upsilon \partial_{\Upsilon} F(Y, \Upsilon)\right), \tag{3.54}$$

which are independently preserved by symplectic transformations. In general, however, to any linear map acting on the $Y^A$ variables one may associate many different symplectic matrices. The particular one defined by (3.52) can be deduced by asking for the generalized prepotential corresponding to the transformed period vector $(C\mathbf{\Pi})'$ to be a *symplectic invariant*, since from (3.45) it follows that

$$\begin{aligned} F'(Y^{A\prime}, \Upsilon') &= F(Y^A, \Upsilon) + 3\frac{\gamma^a D_{abc} Y^b Y^c}{p^0} + 3\frac{\gamma^a \gamma^b D_{abc} Y^0 Y^c}{(p^0)^2} + \frac{D_{abc} \gamma^a \gamma^b \gamma^c (Y^0)^2}{(p^0)^3} \\ &= \frac{D_{abc} Y^{a\prime} Y^{b\prime} Y^{c\prime}}{Y^{0\prime}} + d_a \frac{Y^{a\prime}}{Y^{0\prime}} \Upsilon' + G(Y^{0\prime}, \Upsilon') - d_a \frac{\gamma^a}{p^0} \Upsilon' = F(Y^{A\prime}), \end{aligned} \tag{3.55}$$

thus preserving the original form of the prepotential up to an irrelevant linear term in $\Upsilon$. The main advantage of choosing (3.52) over other possible symplectic maps sharing the same transformation law for the periods $Y^A$ is that $F'(Y^{A\prime}, \Upsilon') = F(Y^{A\prime})$ ensures that the form of the attractor equations is also maintained, as well as that of the entropy formula (cf. (3.51)), whereas in general this may not be the case.

In the next two subsections we will illustrate this point by making explicit use of the symplectic transformation introduced above so as to extend the results of [26] as well as to reduce the general D0-D2-D4-D6 entropy to a simpler one which may be more tractable for studying certain properties exhibited by the relevant quantum corrections studied herein.

### 3.3.1 The general $\mathrm{Re}\,Y^0 = 0$ black hole system

In the following, we aim to extend the analysis of [26, Section 4] to accommodate more general black hole solutions exhibiting a purely imaginary parameter $\alpha$ (cf. eq. (3.9)). However, instead of studying those in full generality, our approach will be to employ the symplectic transformation described in eqs. (3.46)-(3.48) above so as to reduce the corresponding 3-charge problem to an equivalent one with just two independent charges. Furthermore, as it turns out, the simplest such configuration is the D2-D6 solution studied in [26].

Notice that the main advantage of selecting this family of black holes rests on the fact that, regardless of whether one can find a physical solution for the corresponding set of charges, the quantum attractor equations involve contributions with genus no higher than 1, thus remaining algebraic and much more tractable than its more general counterpart. Indeed,

the higher-genus corrections enter explicitly in the attractor relations only via the quantity $\operatorname{Im} G_0$, as can be readily seen from eqs. (3.11)-(3.15). Therefore, the simplest instance to get rid of those is to require $\operatorname{Re} Y^0 = 0$ at the stabilization locus [26]. Imposing this restriction on the attractor equations, one finds

$$q_a = -\frac{1}{p^0} 3D_{abc}(2\operatorname{Re} Y^b)(2\operatorname{Re} Y^c) + \frac{1}{p^0} 3D_{abc}p^b p^c - \frac{4}{p^0} d_a \Upsilon \,, \tag{3.56}$$

$$q_0 = \frac{1}{(p^0)^2} 3D_{abc}p^a(2\operatorname{Re} Y^b)(2\operatorname{Re} Y^c) - \frac{1}{(p^0)^2} D_{abc}p^a p^b p^c + \frac{4}{(p^0)^2} d_a p^a \Upsilon \,, \tag{3.57}$$

which may be combined into the following charge constraint

$$q_0 = -\frac{p^a q_a}{p^0} + \frac{2}{(p^0)^2} D_{abc}p^a p^b p^c \,. \tag{3.58}$$

Notice that this precisely matches eq. (5.22) in [24] with $\operatorname{Im} G_0$ set to zero, where this class of solutions were partially studied. Here we show that this more general configuration can be easily retrieved upon applying the symplectic map discussed in the previous section to the simpler D2-D6 system. Thus, specializing to the latter case by setting $p^a = q_0 = 0$ and using eq. (3.48), we get a new equivalent BPS black hole with charges

$$p^{0\prime} = p^0 \,, \qquad p^{a\prime} = \gamma^a \,, \qquad q_a' = q_a + 3D_{abc}\frac{\gamma^b \gamma^c}{p^0} \,, \qquad q_0' = -\frac{\gamma^a q_a}{p^0} - D_{abc}\frac{\gamma^a \gamma^b \gamma^c}{(p^0)^2} \,. \tag{3.59}$$

These can be readily seen to satisfy the constraint

$$q_0' = -\frac{p^{a\prime} q_a'}{p^{0\prime}} + \frac{2}{(p^{0\prime})^2} D_{abc}p^{a\prime} p^{b\prime} p^{c\prime} \,, \tag{3.60}$$

which exactly coincides with (3.58) above.

Several comments are now in order. First of all, one may easily verify that, upon repeating the same exercise starting with $p^a \neq 0$, the map (3.48) leads us again to the relation (3.60), thereby implying that further symplectic transformations of the form (3.46) still leave us within the same family of solutions. This follows from the fact that they form a subgroup of symmetries, i.e., they are symplectic transformations that preserve the functional form of the generalized prepotential $F(Y^A, \Upsilon)$ (see discussion around eq. (3.55)). On the other hand, from this discussion one concludes that the same analysis performed in [26] regarding the D2-D6 system holds more generally in the presence of D0- and D4-charges provided $\operatorname{Re} Y^0 = 0$ is satisfied at the attractor locus. Consequently, the entropy of any such black hole with $\mathbf{q} = (p^0, p^a, q_0, q_a)^{\mathrm{T}}$ can be directly determined from that associated to an equivalent D2-D6 system with charges

$$p^{0\prime} = p^0 \,, \qquad q_a' = q_a - 3D_{abc}\frac{p^b p^c}{p^0} \,. \tag{3.61}$$

### 3.3.2 The D0-D2-D6 solution

As a second illustration of the symplectic trick described in this section, we consider in what follows BPS black hole solutions whose only constraint comes from not having D4-brane charge. We first derive the corresponding quantum entropy by solving (implicitly) the attractor equations and subsequently show how the general entropy formula (3.2) may be directly obtained from the former by means of the transformation (3.52).

Setting all D4-brane charges to zero results, as per (3.10c), in the rescaled moduli $Y^a$ being purely real. As a consequence, the attractor equations (3.11) simplify to

$$D_{abc}x^a x^b = \Delta_a = -|Y^0|^2 \frac{\tilde{q}_a}{3p^0}, \qquad \tilde{q}_a \equiv q_a + p^0 \frac{d_a \Upsilon}{|Y^0|^2}, \tag{3.62}$$

with $\mathrm{Re}\, Y^a = x^a$, whilst eq. (3.14) reads

$$q_0 - 2\mathrm{Im}\, G_0 = \left[ \frac{2}{3p^0}|Y^0|^2 q_a x^a - \frac{4}{3} d_a x^a \Upsilon \right] \mathrm{Im}\left( (Y^0)^{-2} \right) = \left[ -\frac{2q_a x^a}{3|Y^0|^2} + \frac{4p^0 d_a x^a \Upsilon}{3|Y^0|^4} \right] \mathrm{Re}\, Y^0. \tag{3.63}$$

From here, one may readily determine the black hole entropy to be (cf. eq. (3.17))

$$\begin{aligned} \frac{\mathcal{S}_{\mathrm{BH}}}{4\pi} &= 2\mathrm{Im}\left( \frac{1}{Y^0} \right) \left[ \frac{1}{3p^0}|Y^0|^2 q_a x^a + \frac{1}{3} d_a x^a \Upsilon \right] + \mathrm{Im}\, G - \frac{1}{2} q_0 \mathrm{Re}\, Y^0 \\ &= -\frac{\tilde{q}_a x^a}{3} + \mathrm{Im}\, G - \frac{1}{2} q_0 \mathrm{Re}\, Y^0, \end{aligned} \tag{3.64}$$

where one should substitute the values of $\{x^a, Y^0\}$ solving the attractor eqs. (3.62)-(3.63).

Finally, let us show explicitly how, upon applying the symplectic map (3.48) with $\gamma^a = p^{a\prime}$ to the present black hole system—such that not all $p^{a\prime}$ are vanishing, one recovers the general D0-D2-D4-D6 entropy formula derived in Section 3.2. Indeed, performing the aforementioned transformation results in the old/new gauge charges being related as

$$p^0 = p^{0\prime}, \qquad q_a = q_a' - 3D_{abc}\frac{p^{b\prime}p^{c\prime}}{p^{0\prime}}, \qquad q_0 = q_0' + \frac{p^{a\prime}q_a'}{p^{0\prime}} - 2D_{abc}\frac{p^{a\prime}p^{b\prime}p^{c\prime}}{(p^{0\prime})^2}. \tag{3.65}$$

This allows us to obtain, in particular, the following identifications (cf. eqs. (3.12) and (3.13))

$$\tilde{q}_a = \tilde{q}_a' - 3D_{abc}\frac{p^{b\prime}p^{c\prime}}{p^{0\prime}} = -3p^{0\prime}\frac{\Delta_a'}{|Y^{0\prime}|^2}, \qquad x^a = x^{a\prime} = \mathrm{Re}\, Y^{a\prime} - \frac{\mathrm{Re}\, Y^{0\prime}}{p^{0\prime}}p^{a\prime}, \qquad Y^0 = Y^{0\prime}, \tag{3.66}$$

which converts (3.62) and (3.63) into their more general analogues, i.e., eqs. (3.11) and (3.14), respectively. Furthermore, inserting (3.66) back into eq. (3.64) yields

$$\frac{\mathcal{S}_{\mathrm{BH}}}{4\pi} = p^{0\prime}\frac{\Delta_a'}{|Y^{0\prime}|^2}\mathrm{Re}\, Y^{a\prime} + \mathrm{Im}\, G' - \frac{1}{2}q_0'\mathrm{Re}\, Y^{0\prime} + \frac{1}{2}\frac{\mathrm{Re}\, Y^{0\prime}}{|Y^{0\prime}|^2}p^{a\prime}\left( -\frac{q_a'|Y^{0\prime}|^2}{6p^{0\prime}} + \frac{2}{3}d_a \Upsilon' \right), \tag{3.67}$$

thus reproducing precisely the general entropy formula (3.17), as promised before.

# 4 A Closer Look at Non-Perturbative D0-brane Effects

Our aim in this chapter will be to build on the general results obtained in Section 3 so as to answer some questions that were originally raised in [26]. In particular, we will be interested in understanding whether non-perturbative D0-brane effects can affect the quantum-corrected black hole entropy formula, cf. eq. (3.2). The motivation for this stems from the observation that neither the D0-D2-D4 nor the D2-D6 black hole systems exhibited this class of corrections. In fact, in the former case there were non-trivial contributions to the generalized prepotential found which were associated to the Schwinger poles in the D0-brane one-loop integral. This is in contrast to what happens in the latter system, where those were shown to be absent. However, due to the the different complex phase associated to the non-perturbative residues in the D0-D2-D4 background, there was in the end no physical effect of this kind whatsoever. Hence, one might ask whether this state of affairs will also persist in the most general case, or rather if there is some interesting physics hiding behind this question.

Our strategy to address this issue will consist in using the symplectic symmetry discussed in Section 3.3 so as to reduce the problem to the simpler D0-D2-D6 system, where the formulae becomes easier to handle. We discuss this in Section 4.1, where it is shown that these effects seem to be unavoidably present, in general. Accordingly, this conclusion poses the question as to why the two black hole systems studied in [26] were lacking this type of corrections. To answer this, we revisit in Section 4.2 the semiclassical motion of probe particles in the $\text{AdS}_2 \times \mathbf{S}^2$ near-horizon geometry, both from a wordline as well as algebraic perspectives. We show that BPS states perceive the black hole background in a way such that the electric and magnetic fields effectively felt by those satisfy a quadratic constraint together with their mass (in units of the AdS radius). This implies, in turn, that these particles can never be superextremal, thereby preventing them from escaping and discharging the supersymmetric black hole, even via non-perturbative Schwinger pair production. Finally, in Section 4.3 we briefly comment on the most immediate consequences that follow from this analysis, focusing on a semiquantative understanding of some of the features exhibited by the non-perturbative corrections to the black hole entropy previously derived.

## 4.1 Non-perturbative corrections in the D0-D2-D6 black hole system

Up to this point, we have focused our attention on the quantum deformations of BPS black hole solutions associated to the leading-order *perturbative* effects arising from loop corrections due to D0-brane bound states. However, as is well-known, from the perspective of physical and topological string theory [94, 95] one should expect not only those type of corrections to arise in (3.1), but also further *non-perturbative* contributions. These may also have non-trivial implications for certain black hole observables, such as the BPS quantum entropy [15, 96, 97]. In a previous work [26], it was investigated using two representative examples how the most dominant—at large volume—perturbative effects can affect the entropy of different black hole configurations. The key insight was the realization that the relevant asymptotic series

of corrections could be recast as a Schwinger-like line integral of the form[12]

$$G(Y^0, \Upsilon) = \frac{i}{4(2\pi)^3} \, \chi_E(X_3) \, (Y^0)^2 \, \alpha^2 \oint \frac{ds}{s} \frac{1}{1 - e^{-2\pi i s}} \frac{1}{\sinh^2\left(\frac{\pi \alpha s}{2}\right)} \, . \tag{4.1}$$

For the D0-D2-D4 system it was shown that, even though non-perturbative corrections arising from massive D0-branes actually modify the form of the prepotential, they do not end up contributing to any of the black hole observables, including the entropy index. Additionally, and perhaps somewhat surprisingly, it was found therein that in the case of the D2-D6 black hole configuration, potential non-perturbative contributions appearing in (4.1) were not even present at the level of the prepotential. For those systems, the standard Cauchy prescription for evaluating the above contour integral appears to fail,[13] and a more careful computation was carried out so as to show that the Gopakumar-Vafa representation does not yield any such effects. This nicely matches with the fact that the starting series is Borel summable and thus could be resummed in an exact way without ever having to worry about non-perturbative ambiguities. Note that these two examples correspond to special choices of the expansion parameter $\alpha$ defined in (3.9), being thus purely real or imaginary, respectively.

In contrast, for a generic D0-D2-D4-D6 system, the parameter $\alpha$ takes a complex form, namely $\alpha = |\alpha|e^{i\theta_\alpha}$. Therefore, one may wonder if non-perturbative corrections will contribute to the entropy index in the most general situation. Notice that, from the viewpoint of the Schwinger integral (4.1), a complex-valued $\alpha$ modifies the pole structure of the integrand. Indeed, the non-perturbative poles are now rotated in the complex $s$-plane by an angle $\theta_\alpha$ with respect to the $\alpha \in \mathbb{R}$ case. Specifically, they are found to be located along the ray

$$s = \frac{2\pi n}{|\alpha|} e^{i\left(\frac{\pi}{2} - \theta_\alpha\right)}, \qquad \text{with} \quad n \in \mathbb{Z}, \tag{4.2}$$

whilst the perturbative singularities—i.e., those responsible for providing the exact resummed version of the asymptotic series (3.8)—remain situated along the real $s$-axis, cf. Figure 1, yielding a contribution of the form

$$G^{(p)}(Y^0, \Upsilon) = -\frac{i}{2(2\pi)^3} \, \chi_E(X_3) \, (Y^0)^2 \, \alpha^2 \sum_{n=1}^{\infty} n \log\left(1 - e^{-\alpha n}\right) . \tag{4.3}$$

However, to address this issue we will take advantage of the fact that, via the symplectic transformation (3.52), one may relate the general D0-D2-D4-D6 black hole system to a simpler one without D4-brane charge having the same value of $\alpha$ (see Section 3.3.2 for details). Therefore, starting from the integral (4.1) and assuming without any loss of generality that

---

[12]We remark that a similar and independent derivation of the full contribution to the generalized prepotential due to a given D0-D2 bound state was recently proposed in [35, 36].

[13]It is worth noting that a similar pathological behavior in the Gopakumar-Vafa computation arises whenever the topological string coupling, which is related to our $\alpha$-parameter as $\lambda = 2\pi i \alpha$, becomes purely real [98].

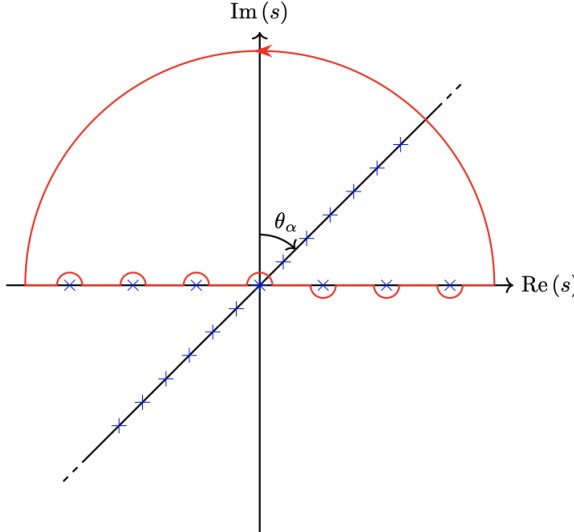

**Figure 1**: Integration contour in the complex s-plane employed to evaluate the loop integral (4.1). The non-perturbative singularities lie along (4.2), whereas the perturbative ones fall onto the real axis. In the limit $\mathrm{Re}\,\alpha \to 0$, all the poles become real.

$\mathrm{Re}\,Y^0 > 0$, one finds a non-perturbative contribution to $G(Y^0, \Upsilon)$ of the form [26]

$$
\begin{aligned}
G^{(np)}(Y^0, \Upsilon = -64) &= \frac{\chi_E}{8\pi^2} Y^0 \sum_{n,k=1}^{\infty} \frac{n}{k} e^{-4\pi^2 k n Y^0} \left(1 + \frac{1}{4\pi^2 k n Y^0}\right) \\
&= \frac{\chi_E}{8\pi^2} Y^0 \sum_{n}^{\infty} \left(n\,\mathrm{Li}_1\left(e^{-4\pi^2 n Y^0}\right) + \frac{1}{4\pi^2 Y^0}\,\mathrm{Li}_2\left(e^{-4\pi^2 n Y^0}\right)\right),
\end{aligned}
\tag{4.4}
$$

such that

$$
\begin{aligned}
G_0^{(np)}(Y^0, \Upsilon = -64) &= -\frac{\chi_E}{2} Y^0 \sum_{n,k=1}^{\infty} n^2 \, e^{-4\pi^2 k n Y^0} = \frac{\chi_E}{2} Y^0 \sum_{n=1}^{\infty} n^2 \frac{1}{1 - e^{4\pi^2 n Y^0}} \\
&= -\frac{\chi_E}{2} Y^0 \sum_{n=1}^{\infty} n^2 \,\mathrm{Li}_0\left(e^{-4\pi^2 n Y^0}\right).
\end{aligned}
\tag{4.5}
$$

On the other hand, the black hole entropy in the D0-D2-D6 case was determined to be

$$
\frac{\mathcal{S}_{\mathrm{BH}}}{4\pi} = -\frac{\tilde{q}_a x^a}{3} + \mathrm{Im}\,G - \frac{1}{2} q_0 \,\mathrm{Re}\,Y^0,
\tag{4.6}
$$

which depends both explicitly on $\mathrm{Im}\,G$, and implicitly on $\mathrm{Im}\,G_0$ via the attactor equations (3.62)-(3.63). Hence, taking the corresponding imaginary parts of (4.4) and (4.5), one obtains

$$\text{Im}\,G^{(np)} = \frac{\chi_E}{8\pi^2} \sum_{n,k=1}^{\infty} \frac{n}{k}\, e^{-4\pi^2 kn \text{Re}\, Y^0} \left[ \frac{p^0}{2} \cos\left(2\pi^2 knp^0\right) - \left(\text{Re}\, Y^0 + \frac{1}{4\pi^2 kn}\right) \sin\left(2\pi^2 knp^0\right) \right],$$

$$\text{(4.7a)}$$

$$\text{Im}\,G_0^{(np)} = -\frac{\chi_E}{2} \sum_{n,k=1}^{\infty} n^2\, e^{-4\pi^2 kn \text{Re}\, Y^0} \left[ \frac{p^0}{2} \cos\left(2\pi^2 knp^0\right) - \text{Re}\, Y^0 \sin\left(2\pi^2 knp^0\right) \right], \qquad \text{(4.7b)}$$

being both non-vanishing, in general. This means, in turn, that they will in principle affect the attractor solution and the black hole entropy for the most general D0-D2-D4-D6 system.

At the same time, this discussion reveals that the absence of non-perturbative D0-brane effects in the D0-D2-D4 and D2-D6 black holes studied in detail in [26] is, in fact, the exception rather than the rule. This observation calls for a proper physical explanation, an issue to which we devote the remainder of our efforts.

## 4.2 Geodesic motion in the near-horizon black hole geometry

To understand why there appear to be cases where non-perturbative corrections to the quantum entropy formula (2.18) are absent for certain BPS states in the theory, it is necessary to fully grasp how the black hole background is perceived by the latter. Only then we may hope to find what distinguishes those systems from the most general situation, where these effects seem to be generically present. To accomplish this, a good starting point would be to study the semiclassical dynamics of probe BPS particles in the two-derivative black hole geometry. This is the aim of this and subsequent sections.

Before proceeding, let us mention that even though a more robust analysis would require to perform the exact quantum path integral associated to these massive fields in $\text{AdS}_2 \times \mathbf{S}^2$, many of the most relevant features can already be understood from a semiclassical point of view, which is our main focus in here. The former challenge will be the subject of an upcoming publication [99], where some of the observations and predictions made in here will be confirmed by solving the full quantum problem in an analogous way to what [28, 29] did.

### 4.2.1 The $\text{AdS}_2 \times \mathbf{S}^2$ spacetime metric

As is well-known, asymptotically-flat, supersymmetric black hole solutions can be regarded as interpolating solitons between two maximally symmetric backgrounds, namely 4d Minkowski at infinity and a Bertotti-Robinson Universe of topology $\text{AdS}_2 \times \mathbf{S}^2$ [40, 41]. The latter occurs close to the horizon, and its physical parameters are completely determined by the black hole charges. Hence, given the universality of the near-horizon geometry, in what follows we will choose the solutions to be moreover double extremal [100], for simplicity. This means that the asympotic vevs of the scalar moduli in the vector-multiplet sector are fixed to their attractor values, which in turn implies that they do not run along spacetime, thereby allowing us to write down explicitly the corresponding line element

$$ds^2 = -f(r)^2 dt^2 + f(r)^{-2} dr^2 + r^2 d\Omega_2^2, \qquad \text{with} \ \ f(r) = 1 - \frac{r_h}{r}. \qquad \text{(4.8)}$$

The above metric coincides with that of an extremal Reissner-Nordström black hole, whose horizon area is given by $4\pi r_h^2$, cf. eq. (2.16). In order to study more closely the near-horizon geometry, we can define a new radial coordinate $y = r - r_h$, which transforms the metric into its familiar isotropic form, namely $f(y) = (1 + \frac{r_h}{y})^{-1}$. Taking the near-horizon limit, $y/r_h \sim 0$, yields the following line element

$$ds^2 = -\frac{y^2}{r_h^2}dt^2 + \frac{r_h^2}{y^2}dy^2 + r_h^2 d\Omega_2^2 \,, \tag{4.9}$$

which corresponds to a Bertotti-Robinson spacetime of mass $M_{\mathrm{BR}}^2 = r_h^2$ and topology given by $\mathrm{AdS}_2 \times \mathbf{S}^2$ [101–103]. This solution is moreover conformally flat, as can be seen by defining another radial coordinate $\rho = r_h^2/y$,[14] thus providing the new metric tensor

$$ds^2 = \frac{r_h^2}{\rho^2} \left( -dt^2 + d\rho^2 + \rho^2 d\Omega_2^2 \right) \,. \tag{4.11}$$

Notice how the $\mathrm{AdS}_2$ boundary, previously located at $y \to \infty$, now lies at $\rho = 0$. Crucially, however, this coordinate system does not cover the whole four-dimensional spacetime, since it only accounts for part of it—namely $y \geq 0$. In fact, one can define multiple coordinate systems to make this manifest, ultimately allowing us to appreciate the global structure of $\mathrm{AdS}_2 \times \mathbf{S}^2$. For instance, one may embed 2d anti-de Sitter within $\mathbb{R}^{2,1}$, which we parametrize by the global coordinates $(X^0, X^2, X^1)$, via the hyperboloid defined by the surface constraint $-(X^0) - (X^2)^2 + (X^1) = -r_h^2 := -R_{\mathrm{AdS}}^2$. This is easily solved by taking $X^0 = R_{\mathrm{AdS}} \cosh(\chi) \sin(\tau/R_{\mathrm{AdS}})$, $X^2 = R_{\mathrm{AdS}} \cosh(\chi) \cos(\tau/R_{\mathrm{AdS}})$ and $X^1 = R_{\mathrm{AdS}} \sinh(\chi)$.[15] Using these coordinates, eq. (4.11) can be written as

$$ds^2 = -\cosh^2 \chi \, d\tau^2 + R_{\mathrm{AdS}}^2 \left( d\chi^2 + d\Omega_2^2 \right) \,. \tag{4.12}$$

Interestingly, in the case of two-dimensional anti-de Sitter space, the fact that $X^1 \in \mathbb{R}$ implies that there are actually two different timelike boundaries located at $\chi \to \pm\infty$ (cf. Figure 2). This phenomenon is also familiar from our experience with Minkowski space in $d$ dimensions, i.e., $\mathbb{R}^{1,d-1}$, where the Penrose diagram admits additional (null) boundaries in the special case of $d = 2$, given that lightrays can go to null infinity by reaching either $x^1 \to \infty$ or $x^1 \to -\infty$. From the above global set of coordinates one may also define another useful parametrization

$$\psi = \sin^{-1} \left( \frac{1}{\cosh \chi} \right) \,, \quad \tilde{\tau} = \frac{\tau}{R_{\mathrm{AdS}}} \,, \qquad \text{with} \quad \psi \in [0, \pi] \,, \quad \tilde{\tau} \in (-\infty, \infty) \,, \tag{4.13}$$

---

[14]In terms of the global hyperboloid embedding, the conformal coordinates arise by identifying

$$X^0 = \frac{y}{r_h}t = \frac{r_h}{\rho}t \,, \quad X^2 - X^1 = y = \frac{r_h^2}{\rho} \,, \quad X^2 + X^1 = \frac{r_h^2}{y} - \frac{yt^2}{r_h^2} = \rho - \frac{t^2}{\rho} \,, \tag{4.10}$$

where in the last step we used the defining relation between $\rho$ and $y$.

[15]Notice that this yields a compact time coordinate with period equal to $R_{\mathrm{AdS}}$. Hence, to actually connect with 2d anti-de Sitter space, one needs to consider the universal covering space of the hyperboloid, denoted here by $C\mathrm{AdS}_2$, which is obtained by unfolding the time direction, thus making it effectively non-compact.

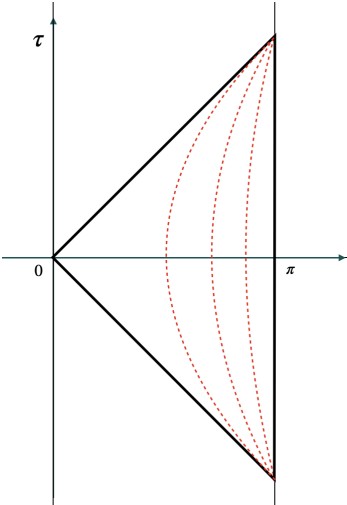

**Figure 2**: Penrose diagrams of AdS spacetimes in different coordinate systems. The triangular region corresponds to that associated to the Poincaré patch, cf. eq. (4.11). The red dotted lines denote constant $\rho$ slices. The global patch, on the other hand, is generated by an infinite array of consecutive Poincaré slices. Notice that the latter includes two boundaries, unlike $\text{AdS}_d$ with $d > 2$.

leading to the following metric tensor[16]

$$ds^2 = \frac{R_{\text{AdS}}^2}{\sin^2 \psi} \left( -d\tilde{\tau}^2 + d\psi^2 \right) + R_{\text{AdS}}^2 d\Omega_2^2 \,, \tag{4.15}$$

in terms of which the two timelike boundaries arise at $\psi = 0, \pi$.

In this section, we are interested in studying the motion of charged (non-)BPS states in a supersymmetric black hole background. In particular, we focus on the near-horizon $\text{AdS}_2 \times \mathbf{S}^2$ geometry (cf. Figure 3), whose metric in Poincaré coordinates is shown in eq. (4.11), with

$$r_h^2 = R_{\text{AdS}}^2 = |C|^2 e^{-K} = |Z_{\text{BH}}|^2 \,. \tag{4.16}$$

On the other hand, any such particle in 4d $\mathcal{N} = 2$ has a mass (in Planck units) given by

$$m = |Z| \,, \qquad Z = e^{K/2} \left( p^A \mathcal{F}_A - q_A X^A \right) \,, \tag{4.17}$$

where $(p^A, q_A)$ denote its magnetic and electric charges. The corresponding worldline action describing its dynamics in the bosonic sector within the Poincaré patch (see Appendix A for a

---

[16]One can easily show that the exact same line element is retrieved upon introducing finite range coordinates in the Poincaré patch (cf. eq. (4.11)) as follows

$$\tilde{\sigma} = \tilde{\rho}_+ + \tilde{\rho}_- \,, \quad \tilde{t} = \tilde{\rho}_+ - \tilde{\rho}_- \,, \qquad \text{with} \quad \tilde{\rho}_\pm = \tan^{-1}(\rho \pm t) \,, \tag{4.14}$$

such that $\tilde{\sigma} \in [0, \pi]$ and $\tilde{t} \in [-\pi, \pi]$. In fact, one finds that the two coordinate patches are precisely identified with one another after taking the universal cover of the time direction $\tilde{t}$.

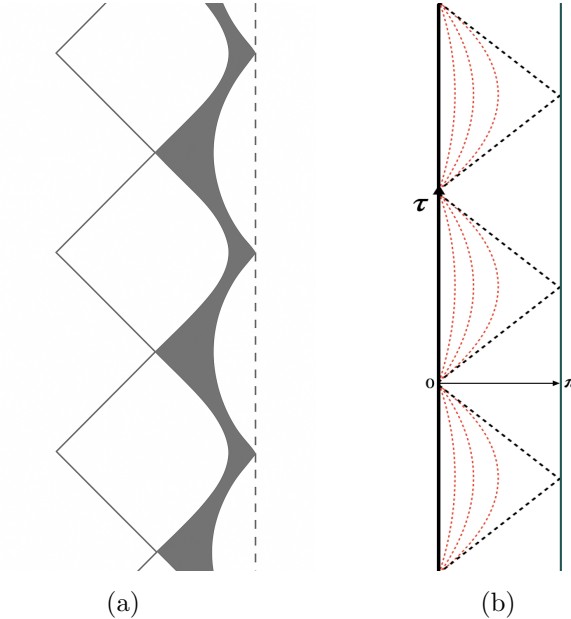

(a)             (b)

**Figure 3**: Penrose diagrams for **(a)** extremal Reissner-Nördstrom black hole and **(b)** global AdS$_2$. The near-horizon region corresponds to the shaded (gray) area in the left image.

similar analysis using global coordinates (4.13)) can be moreover written as follows [43, 104][17]

$$S_{wl} = -2|Z|R_{\mathrm{AdS}} \int_\gamma d\sigma \sqrt{\rho^{-2}\left(\dot{t}^2 - \dot{\rho}^2\right) - \dot{\theta}^2 - \sin^2\theta\,\dot{\phi}^2} + \int_\Sigma p^A G_A - q_A F^A \,, \qquad (4.18)$$

where $\gamma$ denotes the worldline path and $\Sigma$ is any surface embedded in the target spacetime whose boundary precisely coincides with the former, i.e., $\partial\Sigma = \gamma$. Here, we introduced the notation $\dot{x}^\mu := dx^\mu/d\sigma$ and $\sigma$ denotes any convenient worldline parameter. The gauge fields turned on by a black hole of charges $(p^{A\prime}, q'_A)$ in the near-horizon region satisfy the constraints

$$p^{A\prime} = \frac{1}{4\pi}\int_{\mathbf{S}^2} F^A \,, \qquad q'_A = \frac{1}{4\pi}\int_{\mathbf{S}^2} G_A \,, \qquad G_A^- = \bar{\mathcal{N}}_{AB} F^{B,-} \,, \qquad (4.19)$$

which are solved by [42]

$$F^A = R_{\mathrm{AdS}}^{-2}\left(p^{A\prime}\omega_{\mathbf{S}^2} - 2\mathrm{Re}\,CX^A\omega_{\mathrm{AdS}_2}\right)\,, \qquad G_A = R_{\mathrm{AdS}}^{-2}\left(q'_A\omega_{\mathbf{S}^2} - 2\mathrm{Re}\,C\mathcal{F}_A\omega_{\mathrm{AdS}_2}\right)\,, \qquad (4.20)$$

with (cf. eq. (2.17))

$$CX^A = \mathrm{Re}\,CX^A + \frac{i}{2}p^{A\prime}\,, \qquad C\mathcal{F}_A = \mathrm{Re}\,C\mathcal{F}_A + \frac{i}{2}q'_A\,, \qquad (4.21)$$

---

[17]The additional factor of 2 in front of the mass in (4.18) can be deduced by carefully asking for the worldline action to be $\kappa$-supersymmetric [104].

being the stabilized values of the moduli and where we have used the volume 2-forms associated to the $\mathrm{AdS}_2$ and $\mathbf{S}^2$ factors in the above expressions. The latter are given by

$$\omega_{\mathbf{S}^2} = R_{\mathrm{AdS}}^2 \sin\theta d\theta \wedge d\phi \,, \qquad \omega_{\mathrm{AdS}_2} = \frac{R_{\mathrm{AdS}}^2}{\rho^2} dt \wedge d\rho \,, \tag{4.22}$$

and they are moreover related through the four-dimensional Hodge star-operator as follows

$$\star_4 \, \omega_{\mathbf{S}^2} = \omega_{\mathrm{AdS}_2} \,, \qquad \star_4 \, \omega_{\mathrm{AdS}_2} = -\omega_{\mathbf{S}^2} \,, \tag{4.23}$$

as one may easily verify. To actually check that the last condition in (4.19) holds, we first need to determine the anti-self-dual components of the electric and magnetic field strengths. A straightforward calculation reveals, using (4.23), that

$$R_{\mathrm{AdS}}^2 \, F^{A,\,-} = \overline{CX}^A \left( -\omega_{\mathrm{AdS}_2} + i\omega_{\mathbf{S}^2} \right) \,, \qquad R_{\mathrm{AdS}}^2 \, G_A^- = \overline{C\mathcal{F}}_A \left( -\omega_{\mathrm{AdS}_2} + i\omega_{\mathbf{S}^2} \right) \,, \tag{4.24}$$

which indeed satisfy $G_A^- = \bar{\mathcal{N}}_{AB} F^{B,\,-}$, since $\mathcal{F}_A = \mathcal{N}_{AB} X^B$ [47]. One can similarly check that the background (4.20) lies completely along the graviphoton direction, as required by the attractor equations [41, 65, 105]. To show this, we compute the anti-self-dual piece of the graviphoton field strength

$$\begin{aligned} W^- &= e^{K/2}\mathcal{F}_A F^{A,\,-} - e^{K/2} X^A G_A^- = \frac{e^{K/2}}{R_{\mathrm{AdS}}^2} \left[ \mathcal{F}_A \overline{CX}^A - X^A \overline{C\mathcal{F}}_A \right] \left( -\omega_{\mathrm{AdS}_2} + i\omega_{\mathbf{S}^2} \right) \\ &= -ie^{-K/2}\bar{C} R_{\mathrm{AdS}}^{-2} \left( -\omega_{\mathrm{AdS}_2} + i\omega_{\mathbf{S}^2} \right) = Z_{\mathrm{BH}} R_{\mathrm{AdS}}^{-2} \left( \omega_{\mathbf{S}^2} + i\omega_{\mathrm{AdS}_2} \right) \,, \end{aligned} \tag{4.25}$$

thereby implying that the field strength components along all vector multiplet directions exactly vanish, since (see eq. (2.5b))

$$ie^{K/2}\bar{X}^A W^- = \overline{CX}^A R_{\mathrm{AdS}}^{-2} \left( -\omega_{\mathrm{AdS}_2} + i\omega_{\mathbf{S}^2} \right) = F^{A,\,-} \,. \tag{4.26}$$

Let us also note here that, contrary to what happens for the Gopakumar-Vafa derivation in flat space [37], the supersymmetric background turned on by the extremal black hole is neither self-dual nor anti-self-dual, since there is indeed gravitational backreaction and thus the metric is not exactly flat (only conformally so).

With these results at hand, we are finally ready to determine the gauge interaction of the wordline action (4.18). Hence, using (4.20) and (4.21), we find

$$S_{wl} \supset \int_\Sigma p^A G_A - q_A F^A = -q_e \int_\gamma \frac{dt}{\rho} - q_m \int_\gamma \cos\theta d\phi \,, \tag{4.27}$$

with[18]

$$q_e = 2e^{-K/2}\mathrm{Re}\,CZ \,, \qquad q_m = 2e^{-K/2}\mathrm{Im}\,CZ = p^A q_A' - q_A p^{A\prime} \,. \tag{4.28}$$

Notice that the prefactor multiplying the first term in the worldline action can be written as

$$\tilde{m} := 2|Z|R_{\mathrm{AdS}} = 2e^{-K/2}|CZ| \,, \tag{4.29}$$

---

[18]Note that $q_m \in \mathbb{Z}$, in as required by the Schwinger-Swanziger quantization condition [106–108].

after using the explicit expression for the AdS$_2$ radius, cf. eq. (4.16). Therefore, we conclude that for any BPS particle moving in the near-horizon black hole region, we have

$$q_e^2 + q_m^2 = \tilde{m}^2 \implies |q_e| = 2e^{-K/2}\,|\mathrm{Re}\,CZ| \leq 2e^{-K/2}|CZ| = \tilde{m}\,, \tag{4.30}$$

with saturation happening precisely for the extremal case, i.e., whenever $q_m = 0$ holds. This observation will be important later on when discussing non-perturbative effects. More generally, taking also into account non-BPS particles satisfying $m > |Z|$, the condition (4.30) becomes an inequality of the form

$$(2mR_{\mathrm{AdS}})^2 \geq \left|2e^{-K/2}CZ\right|^2 = q_e^2 + q_m^2\,. \tag{4.31}$$

As we will see in upcoming sections, this modification has the effect of enhancing the confining properties that AdS$_2$ imposes on massive states. In what follows, we will solve for the geodesic motion of charged particles in the near-horizon region, as derived from eqs. (4.18) and (4.27). We will do so both from a wordline (Section 4.2.2) and algebraic perspective (Section 4.2.3).

### 4.2.2 Charged geodesics in AdS$_2 \times$ S$^2$: A worldline approach

Let us study first the on-shell trajectories followed by charged massive particles in AdS$_2 \times$ **S**$^2$, obtained by solving the corresponding classical equations of motion. The latter are derived from the action (4.18), which we show here again for convenience

$$S_{wl} = -\tilde{m}\int_\gamma d\sigma \sqrt{\rho^{-2}\left(\dot{t}^2 - \dot{\rho}^2\right) - \dot{\theta}^2 - \sin^2\theta\dot{\phi}^2} - q_e\int_\gamma \frac{dt}{\rho} - q_m\int_\gamma \cos\theta d\phi\,. \tag{4.32}$$

It is useful to rewrite the above functional in an equivalent way using an einbein field $h(\sigma)$

$$S_{wl} = \frac{1}{2}\int_\gamma d\sigma \left[h^{-1}\left(\rho^{-2}\left(-\dot{t}^2 + \dot{\rho}^2\right) + \dot{\theta}^2 + \sin^2\theta\dot{\phi}^2\right) - h\tilde{m}^2\right] - \int_\gamma d\sigma \left(q_e\rho^{-1}\dot{t} + q_m\cos\theta\dot{\phi}\right)\,. \tag{4.33}$$

As usual, the equation of motion associated to the one-dimensional metric, $g_{\sigma\sigma} = h^2(\sigma)$, gives rise to the (Hamiltonian) constraint

$$\rho^{-2}\left(-\dot{t}^2 + \dot{\rho}^2\right) + \dot{\theta}^2 + \sin^2\theta\dot{\phi}^2 + h^2\tilde{m}^2 = 0\,, \tag{4.34}$$

which upon substitution retrieves the original action (4.32). Using now reparametrization invariance in the worldline one can make a gauge choice with $h(\sigma) = 1$, such that the mass-shell constraint can be written as

$$H = \frac{\rho^2}{2}\left[p_\rho^2 - \left(p_t + \frac{q_e}{\rho}\right)^2\right] + \frac{1}{2}p_\theta^2 + \frac{1}{2}\csc^2\theta\,(p_\phi + q_m\cos\theta)^2 \overset{!}{=} -\frac{\tilde{m}^2}{2}\,. \tag{4.35}$$

Above, we have defined the conjugate momenta to the bosonic worldline fields $(t, \rho, \theta, \phi)$

$$p_\rho = \frac{\dot{\rho}}{\rho^2}\,, \quad p_t = -\frac{\dot{t}}{\rho^2} - \frac{q_e}{\rho}\,, \quad p_\theta = \dot{\theta}\,, \quad p_\phi = \sin^2\theta\,\dot{\phi} - q_m\cos\theta\,. \tag{4.36}$$

Notice that, since the Lagrangian does not depend explicitly on $(t, \phi)$, the latter are *ignorable*. Thus, their associated momenta provide conserved quantities of the particle motion, i.e.,

$$\frac{dp_\phi}{d\sigma} = \frac{dp_t}{d\sigma} = 0\,, \tag{4.37}$$

corresponding to the energy and generalized angular momentum (per unit mass), respectively

$$j = p_\phi = \sin^2\theta\,\dot\phi - q_m\cos\theta\,, \quad E = -p_t = \frac{\dot t}{\rho^2} + \frac{q_e}{\rho}\,. \tag{4.38}$$

On the other hand, the equations of motion for the $(\rho, \theta)$-variables are given by

$$\dot p_\rho = \frac{d}{d\sigma}\left(\frac{\dot\rho}{\rho^2}\right) = E\,\frac{\dot t}{\rho} - \frac{\dot\rho^2}{\rho^3}\,, \tag{4.39a}$$

$$\dot p_\theta = \ddot\theta = \sin\theta\left(\cos\theta\,\dot\phi^2 + q_m\dot\phi\right) = \dot\phi\tan\theta\left(\dot\phi - j\right)\,. \tag{4.39b}$$

The latter must be supplemented with the Hamiltonian constraint (4.35), which in terms of the conserved quantities (4.38) reads

$$p_\rho^2 + \frac{p_\theta^2}{\rho^2} + V(\rho, \theta) = 0\,, \qquad V(\rho, \theta) = \frac{\tilde m^2}{\rho^2} - \left(E - \frac{q_e}{\rho}\right)^2 + \frac{1}{\rho^2\sin^2\theta}\left(j + q_m\cos\theta\right)^2\,. \tag{4.40}$$

Notice that, despite the form of eqs. (4.38) and (4.39), the motion along the $\mathrm{AdS}_2$ and $\mathbf{S}^2$ factors do not decouple from each other, since the constraint (4.40) involves both sectors of the worldline Lagrangian, which get intertwined (see also the discussion of Section 4.2.3). Still, a good strategy that will prove useful in the following consists in first solving the sphere dynamics and, subsequently, consider that associated to 2d anti-de Sitter space, taking into account the mass shell constraint. Hence, to obtain the dynamics along the sphere, one may realize that the generalized angular momentum $j$ corresponds to just one component (that along the $x^3$-direction) of a three-dimensional vectorial conserved quantity $\boldsymbol{J}$. The latter can be identified with the total angular momentum of the system, i.e., including that associated to the gauge field. These three quantities provide for generalized conserved charges which render the motion of the particle completely integrable, and corresponding to a uniform precession around the direction determined by the generalized angular momentum vector (see Figure 4). Therefore, proceeding as in the free particle case, if we choose our coordinate system such that the vertical direction is perfectly aligned with $\boldsymbol{J}$, then the charged particle remains for all times at a certain polar angle $\theta$ fixed by the dynamical condition

$$\dot p_\theta = p_\theta = 0 \implies \cos\theta = -\frac{q_m}{j}\,, \qquad \text{with } j = \dot\phi\,, \tag{4.41}$$

hence rotating along the $\phi$-direction with constant angular velocity. Equivalently, this motion may be characterized by exhibiting a precession of the familiar angular momentum vector $\boldsymbol{L}$, whose total length is also constant and equal to $|\boldsymbol{L}| = \ell = \sin\theta\,\dot\phi = \sqrt{j^2 - q_m^2}$.[19]

---

[19]The trajectory described in the main text can also be deduced upon imposing $J_1 = J_2 = 0$ and $J_3 = j$ in (4.51). From there, it follows that $J_\pm = J_1 \pm iJ_2 = \pm ie^{\pm i\phi}\left[p_\theta \pm i\left(\cot\theta\,p_\phi + q_m\csc\theta\right)\right]$ must also vanish,

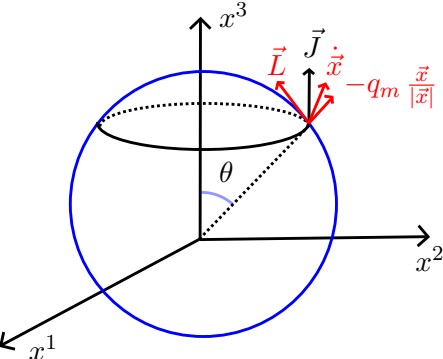

**Figure 4**: Schematic depiction of the geodesic trajectory of a charged particle living on a 2-sphere with a constant and everywhere orthogonal magnetic field. The orbit (black) precesses around a conserved generalized angular momentum vector $\boldsymbol{J} = j\,\partial_z$ at a polar angle fixed by $\cos\theta = -q_m/j$.

Once we have solved for the $\mathbf{S}^2$ part, the dynamics along $\mathrm{AdS}_2$ can be more easily deduced from the Hamiltonian constraint, which leads to the radial equation (imposing $p_\theta = 0$)

$$p_\rho^2 + V(\rho) = 0\,, \qquad V(\rho) = \frac{\tilde{m}^2 + \ell^2}{\rho^2} - \left(E - \frac{q_e}{\rho}\right)^2\,. \tag{4.43}$$

Notice that the presence of a generically non-trivial angular momentum along $\mathbf{S}^2$ amounts to increasing the effective mass of the particle as $\tilde{m}^2 \to m_{\mathrm{eff}}^2 = \tilde{m}^2 + \ell^2$. Hence, we see that the minimum value of $j$ is given by $\pm q_m$ and occurs precisely when the particle is at *rest* in either one of the two poles of the sphere. On the other hand, as we increase the angular momentum, the trajectory is pushed towards $\theta = \pi/2$, reaching the equator in the limit $j = \ell \to \infty$.

**A special class of geodesics**

Let us consider first a special class of solutions that are characterized by having the minimal possible value for the quadratic Casimir on the sphere, namely $j = |q_m|$ or, equivalently, $\ell = 0$ (see Section 4.2.3 for more on this). In this case, the geodesic motion reduces to that in $\mathrm{AdS}_2$, such that from the constraint equation (4.43) one finds

$$p_\rho^2 + V(\rho) = 0\,, \qquad V(\rho) = \frac{\tilde{m}^2}{\rho^2} - \left(E - \frac{q_e}{\rho}\right)^2\,. \tag{4.44}$$

which is depicted in Figure 5 below, and whose solutions yield hyperbolae in the $(t, \rho)$-plane (see discussion around eq. (4.56)). These can be moreover seen to depend crucially on the relative size of the worldline couplings $(\tilde{m}, q_e)$ [109]. For instance, if the electric field is

---

thereby yielding

$$p_\theta = \pm i\,(\cot\theta\,p_\phi + q_m \csc\theta) = 0 \iff \cos\theta = -\frac{q_m}{j}\,. \tag{4.42}$$

such that $q_e^2 < \tilde{m}^2$, the charged particle always remains at a finite distance from the AdS boundary, reaching the Poincaré horizon $\rho \to \infty$ at late times (cf. Figure 5(a)). On the other hand, for large electric fields $q_e^2 > \tilde{m}^2$ the motion changes according to whether $q_e p_t < 0$ or $q_e p_t > 0$. In the former case (cf. Figure 5(b), yellow line), one finds two branches: one staying away from $\rho = 0$—the particle—and reaching eventually the Poincaré horizon; and the other confined near the boundary—the anti-particle, being thus emitted and reabsorbed by the latter. For the opposite relative sign between the charge and energy (cf. Figure 5(b), blue line), the motion covers the entire axis $\rho > 0$, and consists effectively on just one branch. This means that particles are either emitted from the boundary and reach the horizon of the Poincaré patch, or actually come from the latter and are absorbed at the boundary. Finally, when $q_e^2 = \tilde{m}^2$ (cf. Figure 5(c)), the anti-particle trajectory in the two-branch superextremal potential (yellow) disappears into the boundary, whilst that of the particle in the one-branch superextremal analogue (blue) becomes tangent to the latter,[20] see also Figure 6.

Interestingly, for the case at hand it readily follows from (4.30) that the BPS particles exploring the near-horizon geometry of a 4d $\mathcal{N} = 2$ extremal black hole are always *subextremal*, with saturation (i.e., extremality) being possible only when the magnetic charge vanishes. This means that, in general, if those particles get pair produced by quantum fluctuations in the near horizon region, they will remain therein and thus cannot account for any discharge process of the black hole, rendering the BPS solution non-perturbatively stable [99]. The limiting case requires special care, since a priori the particle can reach the boundary of AdS$_2$ in finite global time. However, the fact that it does so at rest is nothing but a manifestation of the no-force condition experienced between the BPS probe and the extremal black hole.

Let us end this discussion by remarking that in the most general situation where the particle precesses around the vertical axis at a constant $\theta$, the qualitative behavior associated to the motion along the AdS$_2$ factor does not get significantly modified. For instance, if $q_e^2 < \tilde{m}^2$ (Figure 5(a)) then the radial equation $p_\rho^2 + V(\rho) = 0$ still prevents charged particles from reaching the boundary of AdS, regardless of their energy and angular momentum. On the other hand, if $q_e^2 > \tilde{m}^2$ then there exists a maximum $\ell$ that the particle can have while still being able to escape the near-horizon region. The extremal situation therefore occurs when $\ell = 0$ and $|q_e| = \tilde{m}$, which we examine next.

**The extremal case**

We consider now the limiting case where the particle is only electrically charged, such that $q_m = 0$ and thus $|q_e| = \tilde{m}$, as seen directly from (4.30). In this setup, the conserved angular momentum along the 2-sphere reduces to the more familiar one, and thus the $\theta$-equation can be solved by conveniently choosing our coordinate system, ensuring that the particle starts at $\theta = \pi/2$ with initial velocity completely aligned along the $\phi$-direction (i.e., $\dot{\theta} = 0$). Consequently, the motion is such that the particle remains at the equator of $\mathbf{S}^2$ at all times

---

[20]This can be easily confirmed by computing $d\rho/dt = \sqrt{1 - \frac{\tilde{m}^2}{(E\rho - q_e)^2}}$, which indeed tends to zero at $\rho = 0$ precisely when $\tilde{m}^2 = q_e^2$.

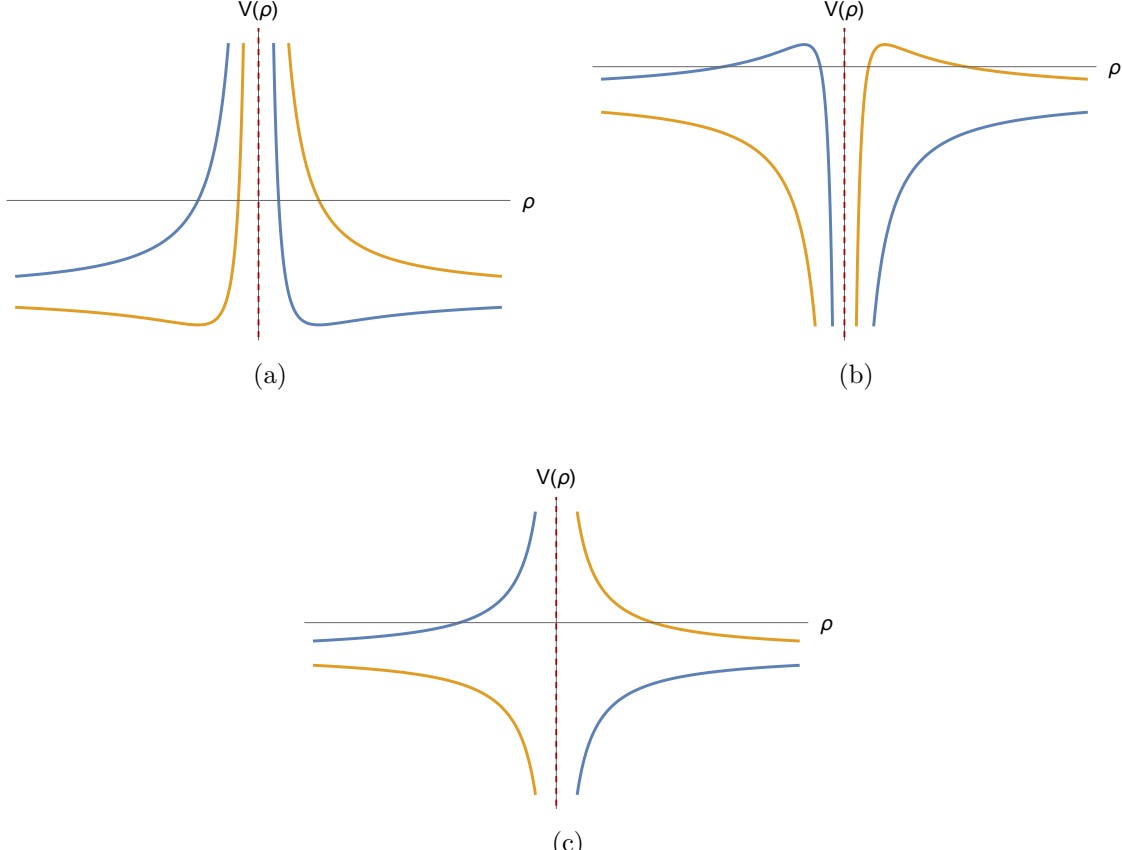

**Figure 5**: Effective scalar potential $V(\rho)$ controlling the radial dynamics in AdS$_2$ after solving the motion along the sphere. A dashed vertical line (red) at $\rho = 0$ denotes the timelike boundary of anti-de Sitter. The qualitative features of the potential depend on whether **(a)** $q_e^2 < \tilde{m}^2 + \ell^2$ (subextremal), **(b)** $q_e^2 > \tilde{m}^2 + \ell^2$ (superextremal), or **(c)** $q_e^2 = \tilde{m}^2 + \ell^2$ (extremal). In each case, we show the corresponding effective potential for both relative signs of the energy $E$ and the electric charge $q_e$, namely for $Eq_e > 0$ (yellow) and $Eq_e < 0$ (blue). Notice that sending $q_e \to -q_e$ with fixed $E$ amounts to the map $\rho \to -\rho$, as can be easily verified from the explicit definition of $V(\rho)$ in eq. (4.43).

and moves within the latter with constant angular velocity given by $\dot{\phi} = j = \ell$. In this case, the radial equation reads

$$p_\rho^2 + V_{\text{ext}}(\rho) = 0 , \qquad V_{\text{ext}}(\rho) = \frac{\tilde{m}^2 + \ell^2}{\rho^2} - \left( E - \frac{\tilde{m}}{\rho} \right)^2 . \tag{4.45}$$

Notice that this brings us back to the situation described around (4.44), where the angular momentum provides an additional positive contribution that enhances the 'confining' effect of the mass in anti-de Sitter space. Indeed, for $\ell \neq 0$ we find ourselves again in the subextremal case (cf. Figure 5(a)), thereby preventing the charged particles from reaching the boundary. On the other hand, if $\ell = 0$ we then recover the expectations from Figure 5(c) in the previous discussion, and the particle can now reach the AdS boundary at threshold.

**The purely magnetic case**

There is another interesting possibility which arises when our particle has only magnetic charge, such that $|q_m| = \tilde{m}$. In this case, the conserved energy reduces to that of a chargeless massive state in AdS$_2$, namely $E = \frac{\dot{t}}{\rho^2}$, and the new Hamiltonian constraint yields

$$p_\rho^2 + V_{\text{mag}}(\rho) = 0 \,, \qquad V_{\text{mag}}(\rho) = \frac{j^2}{\rho^2} - E^2 \,, \tag{4.46}$$

since now $\ell = \sqrt{j^2 - \tilde{m}^2}$. Thus, qualitatively, and regardless of the detailed motion along $\mathbf{S}^2$, the potential $V_{\text{mag}}(\rho)$ exhibits an infinite barrier that prevents the particle from reaching the AdS$_2$ boundary. In order to connect with our general discussion above, it is useful to consider the special class of geodesics with $\ell = 0$. For those, the magnetic potential simplifies again to $V_{\text{mag}}(\rho) = \frac{\tilde{m}^2}{\rho^2} - E^2$, thus recovering the same behavior as in Figure 5(a)—when $q_e \to 0$.

### 4.2.3 Charged geodesics in AdS$_2 \times$ S$^2$: An algebraic approach

Let us remark that, despite the apparent complications arising from introducing the gauge fields along the different subsectors of spacetime, the motion of the particles turned out being completely integrable. This led, ultimately, to very simple trajectories which correspond to circular orbits in $\mathbf{S}^2$—at a fixed polar angle $\theta$—as well as certain (branches of) hyperbolae within AdS$_2$. To explain why this is so, we first note that, since both the electric and magnetic fields are constant and orthogonal to the corresponding 2d surfaces, the physical system inherits the symmetries exhibited by the underlying four-dimensional spacetime [109–111]. These are encoded into the (super-)conformal group $SU(1,1|2)$, which contains $SU(1,1)$ and $SU(2)$ as bosonic subgroups. The first factor indeed corresponds to the conformal isometries of AdS$_2$, generated by $\{K_0, K_\pm\}$, whilst the second one captures the rotational symmetry of the sphere, whose generators we denote in the following by $\{J_0, J_\pm\}$. To see this explicitly, let us consider again the Hamiltonian (4.35), which we recall here for convenience

$$H = \frac{1}{2}\rho^2 \left[ p_\rho^2 - \left( p_t + \frac{q_e}{\rho} \right)^2 \right] + \frac{1}{2}p_\theta^2 + \frac{1}{2}\csc^2\theta \, (p_\phi + q_m \cos\theta)^2 \,. \tag{4.47}$$

In terms of the set of canonical variables $\{q^i\} = \{t, \rho, \theta, \phi\}$ together with their conjugate momenta (cf. eq. (4.36)), which satisfy the usual Heisenberg algebra[21]

$$\left\{ q^j, p_k \right\}_{\text{PB}} = \delta_k^j \,, \tag{4.49}$$

---

[21]The Poisson bracket is defined as

$$\left\{ A(q,p), B(q,p) \right\}_{\text{PB}} = \frac{\partial A}{\partial q^i}\frac{\partial B}{\partial p_i} - \frac{\partial B}{\partial q^i}\frac{\partial A}{\partial p_i} \,. \tag{4.48}$$

Recall that from the Hamilton equations of motion, $\dot{q}^i = \frac{\partial H}{\partial p_i}$, $\dot{p}_i = -\frac{\partial H}{\partial q^i}$, it follows that the (proper) time evolution of any function $A = A(q^i, p_j)$ is determined by its Poisson bracket with the Hamiltonian, namely

$$\frac{dA(q,p)}{d\sigma} = \left\{ A(q,p), H(q,p) \right\}_{\text{PB}} \,.$$

the generators of the corresponding symmetry groups are given (in Chevalley form) by

$$
\begin{aligned}
K_+ &= p_t \,, \\
K_- &= (t^2 + \rho^2)p_t + 2t\rho\, p_\rho + 2q_e\rho \,, \\
K_0 &= t\, p_t + \rho\, p_\rho \,,
\end{aligned}
\tag{4.50}
$$

for the $SL(2, \mathbb{R})$ conformal group of AdS$_2$, and similarly

$$
\begin{aligned}
J_1 &= -\sin\phi\, p_\theta - \cot\theta \cos\phi\, p_\phi - q_m \csc\theta \cos\phi \,, \\
J_2 &= \cos\phi\, p_\theta - \cot\theta \sin\phi\, p_\phi - q_m \csc\theta \sin\phi \,, \\
J_3 &= p_\phi \,,
\end{aligned}
\tag{4.51}
$$

for the rotational $SU(2)$ group associated to the 2-sphere. Note that $K_+$ and $K_0$ have a simple interpretation as (Poincaré) time translation and dilatation rescaling operators, respectively, whereas $K_-$ generates certain non-linear special conformal transformations. With this at hand, one may readily check that these functions satisfy the algebra

$$
\begin{aligned}
\big\{J_i, J_j\big\}_{\mathrm{PB}} &= \epsilon_{ijk} J_k \,, \\
\big\{K_+, K_-\big\}_{\mathrm{PB}} &= -2K_0 \,, \quad \big\{K_0, K_\pm\big\}_{\mathrm{PB}} = \pm K_\pm \,, \\
\big\{J_i, K_j\big\}_{\mathrm{PB}} &= 0 \,,
\end{aligned}
\tag{4.52}
$$

as previously advocated, and moreover that they correspond to conserved quantities of the particle motion, since their commutator with the Hamiltonian vanishes (see footnote 21). Furthermore, it is easy to see that the latter may be written as

$$
2H = \delta^{ik} J_i J_k + K_0^2 - \frac{1}{2}\left(K_+ K_- + K_- K_+\right) - q_e^2 - q_m^2 = C_2^{\mathbf{S}^2} + C_2^{\mathrm{AdS}_2} - \tilde{m}^2 \,,
\tag{4.53}
$$

namely as a sum of two quadratic Casimirs. Therefore, we conclude that the constraint (4.35) can be equivalently recast as the group theoretic restriction

$$
H \overset{!}{=} -\frac{1}{2}\tilde{m}^2 \iff C_2^{\mathbf{S}^2} + C_2^{\mathrm{AdS}_2} = 0 \,,
\tag{4.54}
$$

where the second equality has been obtained by imposing the relation between the particle mass and gauge charges, cf. eq. (4.30). Hence, the classical Hamiltonian constraint simply requires the motion of the particle along the different 2d submanifolds in spacetime to be such that the Casimir elements of the underlying symmetry groups are equal in absolute value and opposite in sign. Notice that for non-BPS particles, one obtains instead

$$
C_2^{\mathbf{S}^2} + C_2^{\mathrm{AdS}_2} = -\Delta^2 \,,
\tag{4.55}
$$

with $\Delta^2 = \tilde{m}^2 - q_e^2 - q_m^2 > 0$, cf. eq. (4.31).

Incidentally, the above algebraic formulation allows us to solve the charged geodesic equations for the AdS$_2$ factor in an implicit way [109]. This follows directly from substituting

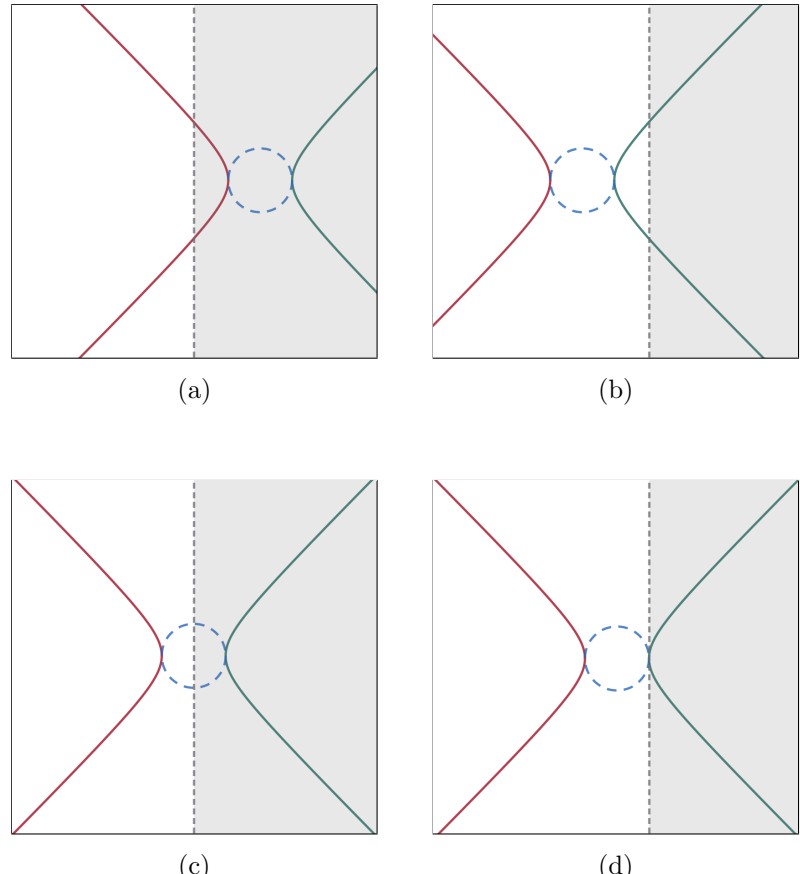

**Figure 6**: Classical paths of charged particles in the Poincaré patch of AdS$_2$ (shaded grey region). The solid green (red) line represents the D0-brane (resp., anti-D0-brane) trajectory parametrized by real proper time, while the dashed blue curves correspond to those in imaginary time. **(a)** For $\bar{m} < |q_e|$, charged particles with $q_e p_t < 0$ can undergo pair production in the bulk. **(b)** In contrast, configurations with $q_e p_t > 0$ do not allow for such effects. **(c)** For $\bar{m} > |q_e|$, the classical trajectories remain confined at a finite radial distance from the boundary, signaling vacuum stability. This corresponds to BPS particles with non-zero magnetic charge, i.e., $q_m \neq 0$. **(d)** Finally, when $\bar{m} = |q_e|$, the anti-D0-brane trajectory asymptotically approaches the AdS boundary and disappears, while the one of the D0-brane becomes tangent to the latter. This special case corresponds to a BPS particle with $q_m = 0$.

$\rho p_\rho = K_0 - t K_+$ and $p_t = K_+$ into (4.53), thus leading to hyperbolic trajectories in Poincaré coordinates of the form

$$\left(\rho + \frac{q_e}{K_+}\right)^2 - \left(t - \frac{K_0}{K_+}\right)^2 = \frac{\tilde{m}^2 + \ell^2}{K_+^2} . \tag{4.56}$$

It is worth emphasizing here that the qualitative behavior associated to the different branches of the classical solutions described after (4.44) is reproduced by choosing accordingly the sign of the $\rho$-coordinate for the center of the hyperbola in (4.56), since $\text{sgn}(q_e/K_+) = \text{sgn}(q_e p_t)$. We have shown this in Figure 6 above.

## 4.3 Summary of the semiclassical picture

To conclude this chapter, we now bring everything together to address the original question posed at the beginning of this section, namely what kind of physics the non-perturbative corrections to the entropy encode, and under what conditions these should be expected to arise. To do so, we will first review and highlight some relevant features that are associated to the euclidean formulation of the classical geodesic problem described above [110], paying special attention to the presence of worldline instanton solutions. The main reason for focusing on euclidean space stems from the fact that, as is by now well-known, certain non-perturbative phenomena—such as Schwinger pair production [112]—may be understood as a quantum tunneling process between classically allowed motions which are nevertheless separated from each other by a finite potential barrier (see e.g., [113–117] for an incomplete list of references). These effects give rise, eventually, to an imaginary part in the field theory action that is responsible for the quantum non-persistence (i.e., decay) of the vacuum under consideration.

We thus begin by writing the euclidean version of the worldline action (4.33), and restrict ourselves to paths which solve the classical equations of motion on the sphere

$$S_E = \frac{1}{2} \int d\sigma \left[ \left( \frac{1}{h\rho^2} \left( \dot{t}_E^2 + \dot{\rho}^2 \right) \right) + h m_{\text{eff}}^2 \right] + q_e \int d\sigma \, \frac{\dot{t}_E}{\rho} \,, \tag{4.57}$$

where we have neglected the purely magnetic coupling since it does not contribute to the integral, and we moreover defined an effective mass of the form $m_{\text{eff}} = \sqrt{\tilde{m}^2 + \ell(\ell+1)}$.[22] Notice that this corresponds to the motion of a charged particle propagating in the hyperbolic plane, which must be supplemented with the on-shell constraint

$$H_E = \frac{\rho^2}{2} \left[ p_\rho^2 + \left( p_{t_E} - \frac{q_e}{\rho} \right)^2 \right] - \frac{m_{\text{eff}}^2}{2} \overset{!}{=} 0 \,, \tag{4.58}$$

where we recall that $p_{t_E}$ provides a conserved quantity associated to $t_E$-translation symmetry. Consequently, the resulting radial equation we are left with reads (cf. eq. (4.43))

$$p_\rho^2 - V(\rho) = 0 \,, \qquad V(\rho) = \frac{m_{\text{eff}}^2}{\rho^2} - \left( p_{t_E} - \frac{q_e}{\rho} \right)^2 \,. \tag{4.59}$$

hence implying that the dynamics is governed now by the *inverse* of the potentials shown in Figure 5. In addition, this means that there might exist classically allowed solutions between the turning points of $V(\rho)$, which can be readily determined to be

$$V(\rho) = 0 \iff \rho_\pm = \frac{q_e \pm m_{\text{eff}}}{p_{t_E}} \,. \tag{4.60}$$

One can also compute analytically the action associated to these closed paths using the fact that the Lagrangian and Hamiltonian of the worldline theory are related via the expression

$$\mathcal{L}_E = p_i \dot{q}^i - H_E = p_i \dot{q}^i \,, \tag{4.61}$$

---

[22] In what follows we impose quantization of angular momentum, which amounts to substituting $\ell^2 \to \ell(\ell+1)$ in the effective potential (4.59) for the radial coordinate in AdS$_2$.

where the second equality exploits (4.58), such that

$$S_E = \oint p_i dq^i \,. \tag{4.62}$$

Let us note that, when specializing to trajectories satisfying the equations of motion derived from eq. (4.57), the integral above reduces to

$$S_E = 2 \int_{\rho_-}^{\rho_+} d\rho \, p_\rho = 2 \int_{\rho_-}^{\rho_+} d\rho \, \sqrt{V(\rho)} \,. \tag{4.63}$$

This follows from the fact that the analogous contributions associated to the coordinates $\{t_E, \theta, \phi\}$ do not modify the classical action since their conjugate momenta are either exactly zero or rather provide some conserved charges, implying that $\oint dq^i = 0$ due to the periodicity of the paths in euclidean space. Therefore, by changing variables to $y = (p_{t_E}\rho - q_e)/m_{\text{eff}}$ one may write the previous integral as

$$S_E = 2m_{\text{eff}} \int_{-1}^{1} dy \, \frac{\sqrt{1-y^2}}{y + q_e/m_{\text{eff}}} \,, \tag{4.64}$$

which, for the case where the electric charge is bigger than the effective mass in units of the AdS radius, yields a finite result equal to

$$S_E = 2\pi \left( q_e - \sqrt{q_e^2 - \tilde{m}^2 - \ell(\ell+1)} \right) \,, \tag{4.65}$$

thus agreeing with the one derived in [109] when setting the angular momentum to zero.[23] Conversely, when $|q_e| < m_{\text{eff}}$ the (simple) pole of the integrand in (4.64) falls within the integration domain, giving rise to a diverging result for the worldline action. This reflects the fact that the corresponding trajectories become now unbounded, eventually reaching the boundary of $\mathbb{H}^2$. Finally, in the extremal case one similarly obtains a finite answer, namely $S_E = 2\pi q_e$, which may be retrieved directly from eq. (4.65) by taking the limit $|q_e| \to m_{\text{eff}}$.[24]

Geometrically, the closed trajectories described herein correspond to circles in the hyperbolic $(t_E, \rho)$-plane [110]. This can be easily deduced from the differential condition

$$\frac{dt_E}{d\rho} = \pm \frac{q_e}{m_{\text{eff}}} \frac{\frac{p_{t_E}}{q_e} - 1}{\sqrt{1 - \frac{q_e^2}{m_{\text{eff}}^2} \left(1 - \frac{p_{t_E}}{q_e}\rho\right)^2}} \,, \tag{4.66}$$

which follows from the (euclidean) equations of motion

$$\dot{t}_E = q_e \rho \left(\frac{p_{t_E}}{q_e}\rho - 1\right) \,, \qquad \dot{\rho} = \pm m_{\text{eff}} \rho \sqrt{1 - \frac{q_e^2}{m_{\text{eff}}^2}\left(1 - \frac{p_{t_E}}{q_e}\rho\right)^2} \,. \tag{4.67}$$

---

[23]More generally, accounting for instanton solutions with non-vanishing $\ell$, we recover the result of [118].

[24]Notice that the integral (4.64) is finite when $|q_e| = m_{\text{eff}}$ despite the fact that the pole of the integrand coincides in that case with one of the boundaries of the integration domain.

The solution to (4.66) reads

$$t_E = t_{E,0} \mp m_{\text{eff}}^2 \frac{\sqrt{1 - \frac{q_e^2}{m_{\text{eff}}^2}\left(1 - \frac{p_{t_E}}{q_e}\rho\right)^2}}{p_{t_E} q_e} \implies (t_E - t_{E,0})^2 + \left(\rho - \frac{q_e}{p_{t_E}}\right)^2 = \frac{m_{\text{eff}}^2}{p_{t_E}^2}, \qquad (4.68)$$

and indeed describes a circle of radius $R_0 = m_{\text{eff}}/p_{t_E}$ centered at $(t_{E,0}, \rho_0 = q_e/p_{t_E})$, as advertised. We have depicted the aforementioned trajectories in Figure 6 as circular dashed lines.

The important point here is that, as explained in Section 4.2.2, the quadratic constraint satisfied by the gauge charges associated to $\mathcal{N} = 2$ (non-)BPS particles propagating in the near-horizon region ensures that those are always subextremal. This implies, in turn, the absence of real instanton solutions with finite action (4.63) in the euclidean theory, which could be responsible for mediating any non-perturbative Schwinger-like vacuum decay. Consequently, we find that the present semiclassical analysis is able to provide quantitative evidence in favor of the stability of the supersymmetric $\text{AdS}_2 \times \mathbf{S}^2$ (as well as the underlying black hole) solutions, in agreement with familiar expectations [119–121]. However, it cannot explain by itself neither the origin nor the nature of the non-perturbative corrections to the BPS black hole entropy obtained in Section 3. Nevertheless, one may still argue why the latter ought to be absent in the two special cases studied in [26], namely the D0-D2-D4 and D2-D6 systems. This is what we explain in what follows.

To accomplish this, let us reconsider for the time being the superextremal instanton paths described before. Their precise contribution to the 2d effective action can be obtained by expanding the following one-loop amplitude

$$S_{\text{eff}}^{2d}[A] = -\int_0^\infty \frac{dh}{h} \int \mathcal{D}t_E(h)\mathcal{D}\rho(h)\, e^{-S_E[x^i, A]}, \qquad (4.69)$$

around the semiclassical solution. Doing so one finds (see Appendix B for details)

$$S_{\text{eff}}^{2d}[A] \sim i \int d^2x \sqrt{g} \sum_{k=1}^\infty \sum_{\ell=0}^{\ell_{\text{max}}} f\left(q_e^2 - \tilde{m}^2 - \ell(\ell+1)\right) e^{-2\pi k(q_e - \sqrt{q_e^2 - \tilde{m}^2 - \ell(\ell+1)})} + \ldots, \quad (4.70)$$

when using the saddle point approximation and after summing over all instanton sectors. There are various salient features of the above formula which are worth emphasizing here. First, one realizes that the contribution of superextremal solutions with different values of the angular momentum along the sphere need to be considered. This secretly encodes the fact that the theory is four-dimensional in origin. Second, the prefactor accompanying each of those terms—associated to one-loop fluctuations around the saddle point—exhibits a functional dependence on the quantity $q_e^2 - m_{\text{eff}}^2$ that exactly vanishes at extremality [122].[25] Hence, even if there exist potential non-perturbative instanton solutions in the near-horizon region

---

[25]Notice that this happens when the probe particle and black hole are mutually BPS (in 4d Minkowski), such that they both saturate the extremality bound and thus exert no force on each other [123].

with $|q_e| = \tilde{m}$ yielding a finite action, their overall contribution to the effective Lagrangian switches off. Note that this precisely explains why we did not see any such effect in the D0-D2-D4 background due to D0-branes, since those are exactly extremal in that case [26].

On the other hand, D0-brane probes in the near-horizon geometry sourced by the D2-D6 system (and symplectic duals thereof, see Section 3.3.1) behave as freely moving particles in the AdS$_2$ factor whilst being subject to a magnetic field $B = \tilde{m}/R^2_{\text{AdS}}$ that is everywhere orthogonal to the 2-sphere. Consequently, and despite them belonging to the subextremal case discussed before—where no real instanton contributes to the (imaginary part of the) effective action, the fact that the background becomes purely magnetic in this case also explains the absence of non-perturbative effects associated to the constant graviphoton field strength [124], thus confirming the picture advocated in [26] based on a superficial Borel analysis.

**Non-perturbative corrections via complex saddles?**

In spite of the fact that there seems to be no real euclidean instantons with finite action in the subextremal case that could account for the origin of the non-perturbative corrections to the entropy which are central to this work, it is still valuable to reconsider and have a look at the precise structure exhibited by those, in light of the developments in the preceding sections. Hence, writing the central charge of the D0-probes in units of the AdS$_2$ radius

$$e^{-K/2} C Z_{\text{D0}} = n C X^0 = \frac{1}{2} \left( q_e^{(n)} + i q_m^{(n)} \right) , \tag{4.71}$$

with $n \in \mathbb{Z}$ labeling the (bound) state in the Kaluza-Klein tower, one can write the first term in eq. (4.7a) as follows

$$\text{Im}\, G^{(np)} \supset -\frac{\chi_E}{16\pi^2} \sum_{n,k=1}^{\infty} \frac{q_m^{(n)}}{k} \text{Re} \left[ e^{-2\pi^2 k \left( q_e^{(n)} + i q_m^{(n)} \right)} \right] . \tag{4.72}$$

Notice that a very similar expression than the one thus obtained may be retrieved by performing a naive analytic continuation of the contribution associated to the $\ell = 0$ sector within the 2d effective Lagrangian (4.70), where one needs to use the constraint (4.30) to rewrite $q_e^2 - \tilde{m}^2$ in terms of the magnetic charge $q_m$ and take the determinant resulting from the one-loop fluctuations around the instanton to be proportional to $\sqrt{q_e^2 - \tilde{m}^2}$ (cf. Appendix B for details). In particular, one observes an exponential damping depending on the effective electric field perceived by the BPS probe particle as well as an oscillatory behavior determined by the corresponding magnetic charge. Therefore, given that the structure is very similar to that associated to the (superextremal) worldline instantons above, it is tempting to speculate at this point about the possibility of interpreting these non-perturbative corrections as arising from the contribution of some other complex semiclassical saddle appearing in the 1d quantum-mechanical path integral (4.69). Indeed, it has been argued [125–129] that a correct semiclassical treatment of the quantum theory oftentimes requires from the complexification of the both the action and path integral measure, as well as the inclusion of more general complex saddles contributing to the latter. It would be very interesting to see whether an analysis along these lines could in fact reproduce the same behavior observed in eq. (4.72).

# 5 Conclusions and Outlook

In this work, we have continued the investigation of quantum corrections, both perturbative and non-perturbative, to the entropy of supersymmetric black holes. Our analysis focused on BPS solutions arising in four-dimensional $\mathcal{N} = 2$ effective field theories obtained by compactifying Type IIA string theory on Calabi–Yau threefolds, specializing to the large volume regime. There, the aforementioned quantum effects are encoded into certain higher-derivative operators that can be written as integrals over half-superspace, and whose leading-order terms (at large volume) arise from perturbative $\alpha'$-corrections in the string frame or, equivalently, as loop effects due to D0-branes from a dual M-theory perspective [28, 29]. Particularly interesting is the observation, first made in [26], that for certain black hole solutions—namely the D0-D2-D4 and D2-D6 systems—non-perturbative D0-brane effects are absent and thus do not modify the quantum attractor equations nor the black hole entropy. Hence, one of our original motivations was to elucidate whether this state of affairs would persist when considering other configurations with arbitrary gauge charges.

To address this question, we first derived the quantum entropy formula for the most general D0-D2-D4-D6 black hole system in Section 3. The resulting expression is shown in eq. (3.2). This was done following two alternative but ultimately equivalent routes. The first one consisted of a direct computation by (implicitly) solving the quantum-corrected attractor equations, as detailed in Section 3.2. Alternatively, in Section 3.3, we exploited the duality symmetries exhibited by the 4d $\mathcal{N} = 2$ theory to relate the relevant physical observables of seemingly unrelated black holes—i.e., those with different gauge charges—via certain special symplectic transformations. Incidentally, this strategy also allowed us to trivially extend the results of [26] and obtain the solution of the general system having $\mathrm{Re}\,Y^0 = 0$ at the attractor locus, besides the D2-D6 configuration. Subsequently, we were able to show (cf. Section 4.1) that non-perturbative D0-brane effects are generically present and modify the quantum observables of BPS black holes, including their entropy function. Here, by non-perturbative we mean from the perspective of the dual Schwinger-like Gopakumar-Vafa representation [95] or, similarly, with respect to the coupling constant $\lambda$ of an auxiliary topological string theory [71–73]. Hence, we concluded that the absence of such corrections in the D0-D2-D4 and D2-D6 systems studied in [26] was a particular feature of those, which was hence asking for a proper physical explanation.

Consequently, in Section 4.2 we initiated our investigations to uncover the physical origin of the aforementioned non-perturbative corrections to the black hole entropy. To do so, we adopted a semiclassical approach, where the attention was placed on understanding the dynamics of BPS probe particles in the (near-horizon) black hole background. The focus on the near-horizon region is justified by the attractor mechanism [40, 65–67], which ensures that all scalar moduli flow to fixed values at the horizon determined solely by the black hole charges. Therefore, the resulting geometry arises universally and completely characterizes the underlying black hole physics. Accordingly, we considered the two-derivative (i.e., classical)

approximation so as to isolate the essential features relevant for the probe dynamics, without the complications arising from backreaction effects. Building on earlier works [42, 104, 109], we were able to show that (non-)BPS dyonic particles are always subextremal in the $\mathrm{AdS}_2 \times \mathbf{S}^2$ near-horizon region, in the sense that their (effective) electric charge is smaller than their mass, measured in units of the AdS radius. This follows from the quadratic constraint satisfied by the electric and magnetic charges of the particle, cf. eq. (4.30), and it implies that the zero-brane probes cannot classically escape the AdS throat of the black hole. On the other hand, from the euclidean perspective, the previous observation translates into not having Schwinger pair production, thus rendering the solutions non-perturbatively stable. Furthermore, this analysis is able to single out the D0-D2-D4 and D2-D6 systems, which are perceived by D0-brane probes as purely electric (and therefore extremal) or purely magnetic, respectively. Notice that this was already suggested in [26] by performing a resurgent analysis of the leading-order perturbative quantum contribution to the generalized Type IIA prepotential. Therefore, even if the semiclassical approach is not fully able to argue for the precise form of the non-perturbative contributions to the black hole entropy in general, it can explain why they should be absent in the two aforementioned special systems. Indeed, in the D0-D2-D4 case one finds a wordline instanton solution with finite action but whose prefactor vanishes identically due to the extremality condition $q_e = \tilde{m}$ [122]. On the contrary, the D2-D6 black hole, being purely magnetic, should never be able to account for non-perturbative Schwinger-like corrections [124]. This agrees superficially with the fact that the former exhibits non-trivial contributions of this kind within the generalized prepotential which nevertheless do not affect the theory due to their complex phase, whereas the latter does not exhibit any such corrections to start with [26].

An intriguing understanding of this phenomenon arises from recognizing that the form of the non-perturbative terms in the generalized prepotential resembles that of the Schwinger amplitude. In the physical Schwinger effect, particle-antiparticle pairs are produced when the electric charge-to-mass ratio exceeds a critical threshold [109, 122], leading to an imaginary contribution to the effective action which signals vacuum decay. Within the black hole background, this would precisely account for a discharge process induced by the superextremal particle. However, in the subextremal case, the contribution of these worldline instanton solutions vanishes due to their infinite action. Still, our analysis in Section 4 shows that there exist in general non-trivial exponentially suppressed corrections to the entropy, which are real and and thus cannot be linked with any instability of the background. Instead, these contributions must be interpreted as quantum corrections to the effective Lagrangian encoding virtual processes that are forbidden in the classical limit. A detailed analysis and physical justification for this interpretation will be presented elsewhere [99].

Our results may open up several promising avenues for future research. First of all, to actually account for the presence and the origin of non-perturbative D0-brane corrections in the most general Calabi–Yau black hole system, a more rigorous path integral computation is needed. Results along this direction will be reported in an upcoming publication [99], where

precise quantitative agreement with the analysis of Section 4.1 is found. Still, one may wonder at this stage whether these effects really go beyond the semiclassical picture or rather they could be somehow captured by the former in some intricate way, similarly to what happens with quantum tunneling [130]. In this regard, and based on a naive analytic extension of the superextremal computation, we pointed out in Section 4.3 that an interesting possibility would be the presence of additional complex saddles in the worldline path integral that could precisely reproduce these corrections [125–129]. An interpretation along these lines could also provide valuable insight into the physical meaning of non-perturbative quantum-gravitational phenomena. A full exploration of these matters is left for the future.

Another promising direction involves the study of non-perturbative quantum effects in the vicinity of other infinite distance singularities in moduli space, beyond the large volume point of Calabi–Yau compactifications that has been the primary focus of this work. Among these, particularly interesting are the so-called emergent string and F-theory limits [81]. There, the dominant tower of light states consists not of D0-branes, but rather of a weakly coupled fundamental string or infinitely many D0-D2 bound states, respectively. Such limits admit a perturbative description in terms of a dual heterotic string theory on K3$\times$$\mathbf{T}^2$ or 6d F-theory compactified on an elliptically fibered Calabi–Yau threefold. Investigating the behavior of non-perturbative corrections in these settings, as well as studying BPS black hole transitions near the aforementioned limits would be of significant interest. These questions are also closely tied to the UV-IR connection between black holes and infinite towers of (light) states [131–145] and to the physics of small black hole solutions [8, 146], where classical supergravity ceases to be fully reliable and genuine quantum gravity effects may even become dominant. We aim to return to these questions in future work.

We hope that our efforts will encourage further investigations into these and related exciting research directions.

## Acknowledgements

We are indebted to José Calderón-Infante, Jinwei Chu, Bernardo Fraiman, Damian van de Heisteeg, Álvaro Herráez, Elias Kiritsis, Puxin Lin, Miguel Montero, Eran Palti, Tomás Ortín, Sav Sethi, Gary Shiu, Max Wiesner and Cumrun Vafa for illuminating discussions and very useful comments on the manuscript. A.C., D.L. and C.M. would like to thank Harvard University and its Swampland Initiative for hosting and providing a stimulating environment where parts of this work were completed. The work of A.C. is supported by a Kadanoff and an Associate KICP fellowships, as well as through the NSF grants PHY-2014195 and PHY-2412985. The work of D.L. is supported by the Origins Excellence Cluster and by the German-Israel-Project (DIP) on Holography and the Swampland. A.C. and M.Z. are also grateful to Teresa Lobo and Miriam Gori for their continuous encouragement and support.

# A  Global Geodesic Analysis in AdS$_2 \times$ S$^2$

The purpose of this appendix is to outline how the analysis of Section 4.2 gets modified if one uses a global chart to cover the full 4d AdS$_2 \times$ S$^2$ spacetime instead of focusing just on the Poincaré patch within anti-de Sitter space. Along the way, we also briefly comment on the possibility of having supersymmetry-preserving trajectories, in connection with [42–44, 147].

To do so, we will employ the so-called strip coordinates $(\tilde{\tau}, \psi)$ in 2d AdS (cf. eq. (4.13)), in terms of which one obtains the following worldline action

$$S_{wl} = -\tilde{m} \int_\gamma d\sigma \sqrt{\csc^2 \psi \left( \dot{\tilde{\tau}}^2 - \dot{\psi}^2 \right) - \dot{\theta}^2 - \sin^2 \theta \dot{\phi}^2} + \int_\Sigma p^A G_A - q_A F^A \,, \tag{A.1}$$

with the gauge fields still given by (4.20), when expressed using the appropriate volume forms of AdS$_2$ and S$^2$. The latter is explicitly shown in (4.22), whereas the former would now be

$$\omega_{\text{AdS}_2} = \frac{R_{\text{AdS}}^2}{\sin^2 \psi} d\tilde{\tau} \wedge d\psi \,, \tag{A.2}$$

thus leading to the following gauge contribution within the action functional

$$S_{wl} \supset \int_\Sigma p^A G_A - q_A F^A = - \int_\gamma d\sigma \left( q_e \frac{\dot{\tilde{\tau}}}{\tan \psi} + q_m \cos \theta \, \dot{\phi} \right) \,, \tag{A.3}$$

where $q_e$ and $q_m$ can be found in eq. (4.28). Hence, to determine the classical trajectories followed by BPS particles we introduce an einbein field $h(\sigma)$, such that (A.1) becomes

$$S_{wl} = \frac{1}{2} \int_\gamma d\sigma \left[ h^{-1} \left( \csc^2 \psi \left( -\dot{\tilde{\tau}}^2 + \dot{\psi}^2 \right) + \dot{\theta}^2 + \sin^2 \theta \dot{\phi}^2 \right) - h \tilde{m}^2 \right] - \int_\gamma d\sigma \left( q_e \frac{\dot{\tilde{\tau}}}{\tan \psi} + q_m \cos \theta \dot{\phi} \right) \,. \tag{A.4}$$

By making use of the reparametrization invariance in the worldline we can set $h(\sigma) = 1$, provided we also enforce the following Hamiltonian constraint

$$H = \frac{\sin^2 \psi}{2} \left[ p_\psi^2 - \left( p_{\tilde{\tau}} + \frac{q_e}{\tan \psi} \right)^2 \right] + \frac{1}{2} p_\theta^2 + \frac{1}{2} \csc^2 \theta \, (p_\phi + q_m \cos \theta)^2 \overset{!}{=} -\frac{\tilde{m}^2}{2} \,, \tag{A.5}$$

which depends explicitly on the conjugate momenta

$$p_\psi = \frac{\dot{\psi}}{\sin^2 \psi} \,, \quad p_{\tilde{\tau}} = -\frac{\dot{\tilde{\tau}}}{\sin^2 \psi} - \frac{q_e}{\tan \psi} \,, \quad p_\theta = \dot{\theta} \,, \quad p_\phi = \sin^2 \theta \, \dot{\phi} - q_m \cos \theta \,. \tag{A.6}$$

The new Noether charges are the angular momentum and (global) energy[26]

$$j = p_\phi \,, \quad \tilde{E} = -p_{\tilde{\tau}} \,, \tag{A.7}$$

---

[26]Notice that from the $SL(2, \mathbb{R})$ algebra in eq. (A.10) below, one finds $p_{\tilde{\tau}} = (K_+ + K_-)/2$, in contrast to the Poincaré energy, which is given instead by $p_t = K_+$, cf. (4.50).

whilst the equations of motion for the $(\psi, \theta)$-coordinates read

$$\dot{p}_\psi = \frac{d}{d\sigma}\left(\frac{\dot\psi}{\sin^2\psi}\right) = \frac{\dot{\tilde\tau}}{\cos\psi\sin\psi}(\tilde E - \dot{\tilde\tau}) - \frac{\dot\psi^2}{\sin^3\psi}\cos\psi\,, \tag{A.8a}$$

$$\dot{p}_\theta = \ddot\theta = \sin\theta\left(\cos\theta\,\dot\phi^2 + q_m\dot\phi\right) = \dot\phi\tan\theta\left(\dot\phi - j\right)\,. \tag{A.8b}$$

These must be supplemented with the Hamiltonian constraint (A.5) which, after solving for the sphere dynamics (see discussion after (4.40)), can be conveniently written as

$$p_\psi^2 + V(\psi) = 0\,, \qquad V(\psi) = \frac{\tilde m^2 + \ell^2}{\sin^2\psi} - \left(\tilde E - \frac{q_e}{\tan\psi}\right)^2\,. \tag{A.9}$$

The profile for the effective radial scalar potential is shown in Figure 7, depending on whether the charge of the particle is greater, equal, or smaller than its effective mass $m_{\text{eff}}^2 = \tilde m^2 + \ell^2$.

**The algebraic formulation in global coordinates**

Similarly to what we did in Poincaré coordinates, one can show that the formulation of the classical motion in an algebraic way also holds when using global coordinates, whereby the generators of $SL(2,\mathbb{R})$ are now given by

$$\begin{aligned} K_0 &= \sin\psi\cos\tilde\tau\,p_\psi + \cos\psi\sin\tilde\tau\,p_{\tilde\tau} - q_e\sin\psi\sin\tilde\tau\,, \\ K_\pm &= \mp\sin\psi\sin\tilde\tau\,p_\psi + (1\pm\cos\psi\cos\tilde\tau)p_{\tilde\tau}\mp q_e\sin\psi\cos\tilde\tau\,. \end{aligned} \tag{A.10}$$

These can be checked to verify the Poisson commutation relations

$$\left\{K_+, K_-\right\}_{\text{PB}} = -2K_0\,, \quad \left\{K_0, K_\pm\right\}_{\text{PB}} = \pm K_\pm\,, \tag{A.11}$$

and they can be moreover used to write down the implicit equation for the AdS$_2$ geodesic trajectories using strip coordinates:

$$(K_+ - K_-)\cos\tilde\tau + 2K_0\sin\tilde\tau = -2q_e\sin\psi + (K_+ + K_-)\cos\psi\,. \tag{A.12}$$

Notice that the qualitative behavior of the trajectory followed by the charged particle depends implicitly on the mass and electric charge via the on-shell constraint satisfied by the hyperbolic conserved quantities (cf. eq. (4.53)), which reads

$$\frac{1}{2}\left(K_+K_- + K_-K_+\right) - K_0^2 = \Delta^2 + j^2\,, \tag{A.13}$$

where $\Delta^2 = \tilde m^2 - q_e^2 - q_m^2$ and $j^2$ corresponds to the Casimir of $SU(2)$ associated to the motion along the 'internal' $\mathbf{S}^2$. Therefore, denoting $K_+ + K_- = c$, $K_+ - K_- = a$ and $2K_0 = b$, one may easily verify that depending on whether $c^2 - a^2 - b^2$ is greater, equal, or smaller than zero one obtains a subextremal, extremal, respectively superextremal trajectory, cf. Figure 8. Similarly, by tuning the relative sign between $q_e$ and $K_+$ one can jump between the different branches of the radial potential $V(\psi)$, effectively exchanging the two asymptotic boundaries.

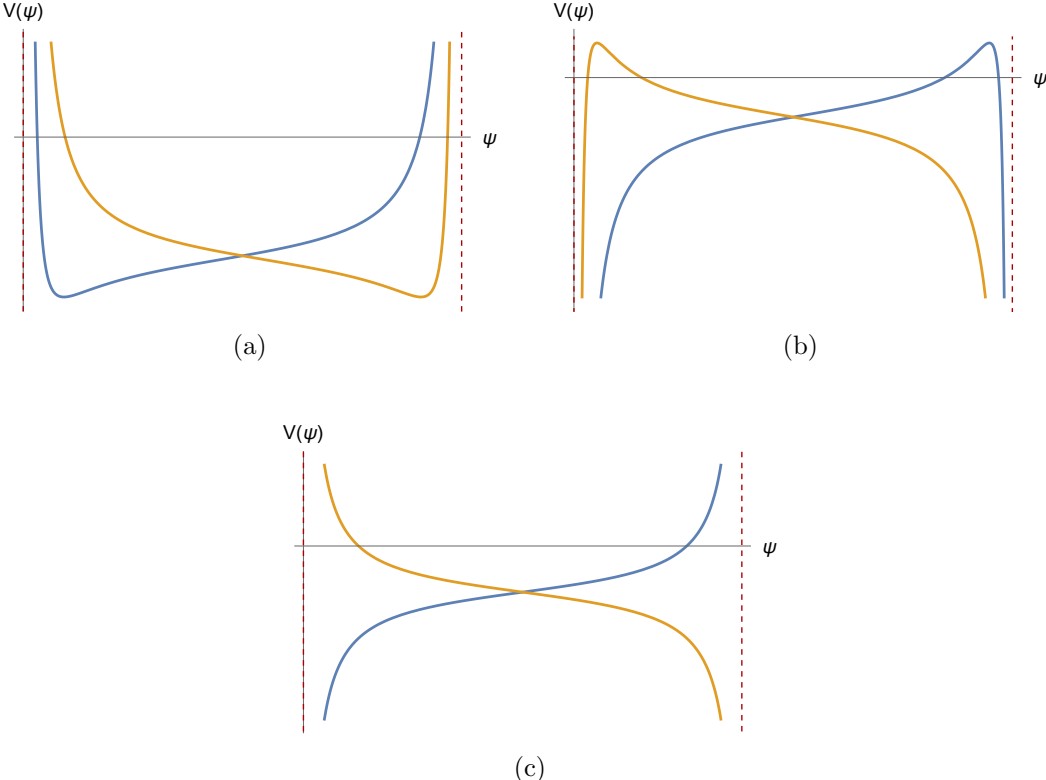

**Figure 7**: Effective scalar potential $V(\psi)$ controlling the radial dynamics in global AdS$_2$ after solving for the motion along the sphere, which amounts to a mass enhancement of the form $\tilde{m}^2 \to \tilde{m}^2 + \ell^2$. The dashed vertical lines denote the two asymptotic boundaries of anti-de Sitter space. The qualitative features of the potential depend on whether **(a)** $q_e^2 < \tilde{m}^2 + \ell^2$ (subextremal), **(b)** $q_e^2 > \tilde{m}^2 + \ell^2$ (superextremal), or **(c)** $q_e^2 = \tilde{m}^2 + \ell^2$ (extremal). In each case, we show the corresponding effective potential for both relative signs of the global energy $\tilde{E}$ and the electric charge $q_e$, namely for $\tilde{E}q_e > 0$ (yellow) and $\tilde{E}q_e < 0$ (blue). Notice that sending $q_e \to -q_e$ for fixed $\tilde{E}$ amounts to the map $\psi \to \pi - \psi$, thus effectively exchanging the two asymptotic timelike boundaries at $\psi = 0, \pi$.

### Supersymmetric trajectories

Lastly, given the supersymmetry exhibited by the background as well as the BPS condition associated to the particles considered in this work, one may wonder whether some of the above trajectories could be supersymmetric. Indeed, it has been recently argued in [148] that all the stationary paths in AdS$_2$—corresponding to special cases of the charged geodesics described by (A.12)—are actually $\frac{1}{2}$-BPS. These include the fully static configurations discussed in [42], which were shown to preserve half of the supersymmetries using a $\kappa$-symmetry argument [104], but also extend the former to accommodate situations with non-trivial angular momentum along the sphere. We illustrate this in the following.

Let us start with the former type of motions. In terms of the global coordinates (4.12), the static paths are characterized by having constant values for $(\chi, \theta, \psi)$, thereby implying

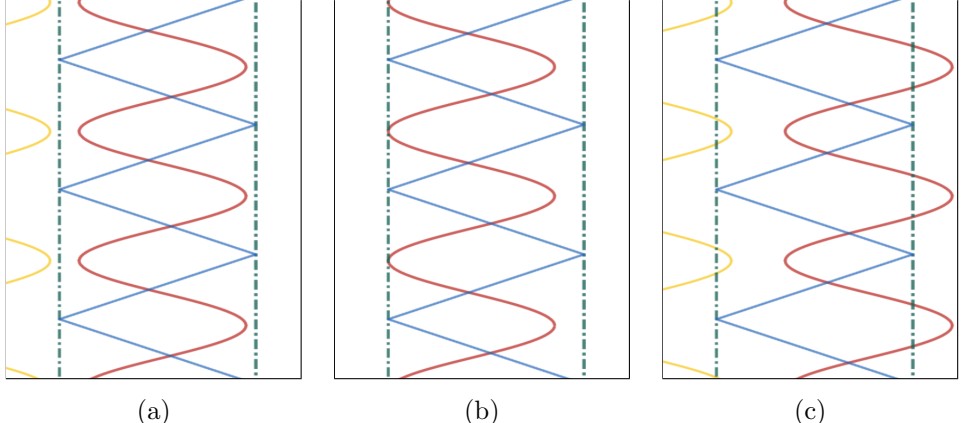

(a)           (b)           (c)

**Figure 8**: Classical trajectories of a charged particle in global AdS$_2$. The dotted green lines denote the boundaries $\psi = 0$, $\pi$, whereas the blue lines correspond to the asymptotic horizon of every Poincaré patch. Finally, the red (yellow) lines depict the D0 (resp. anti-D0) trajectories in real proper time. We show this for each case, namely **(a)** subextremal, **(b)** extremal, and **(c)** superextremal.

that they are associated with the $\ell = 0$ sector. The precise radial distance is fixed to be [42]

$$\tanh \chi = \frac{\operatorname{Re} CZ}{|CZ|} = \frac{q_e}{\tilde{m}}, \tag{A.14}$$

where in the last step we substituted the BPS relation (4.17). Hence, we first ask under what conditions the intrinsic trajectories (A.12) become fully static. This requires to have

$$K_+ = K_-, \qquad K_0 = 0, \qquad \ell = 0, \tag{A.15}$$

thus eliminating any explicit time dependence, both in AdS and $\mathbf{S}^2$. Crucially, due to the subextremality of the BPS particles, the above restriction is seen to be compatible with (A.13), which now reads

$$K_+^2 = q_m^2. \tag{A.16}$$

At the same time, the Hamiltonian constraint forces the particle to be located at a constant radial distance given by[27]

$$\tan \psi = -\frac{\sqrt{q_m^2}}{q_e}. \tag{A.17}$$

Interestingly, this can be seen to coincide exactly with the supersymmetric condition (A.14), when expressed in strip coordinates, since

$$\cos \psi = -\frac{q_e}{\sqrt{q_e^2 + q_m^2}} = -\tanh \chi. \tag{A.18}$$

Hence, we conclude that static paths of the form discussed before satisfy the equations of motion and moreover preserve half of the supersymmetries of the background. This can be

---

[27]Note that there is an implicit sign choice when solving eq. (A.16), which corresponds to swapping the two asymptotic boundaries of AdS$_2$, or equivalently, exchanging particle and anti-particle.

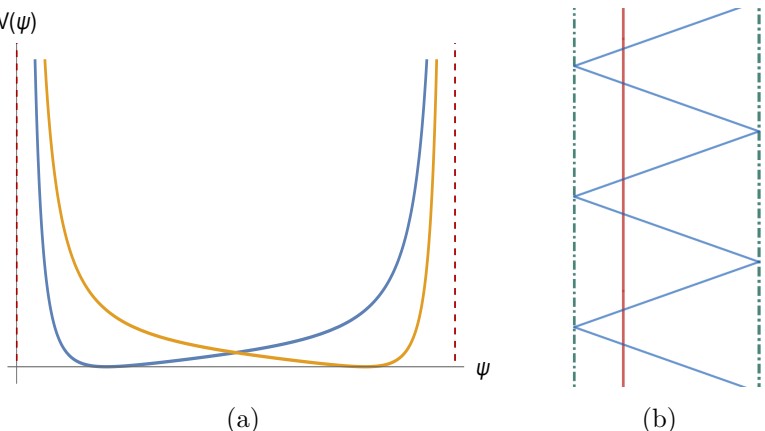

(a)            (b)

**Figure 9**: For supersymmetric trajectories, characterized by having $K_0 = 0$ and $K_+ = K_- = p_{\tilde{\tau}}$, the dynamics of the BPS system corresponds to that of a stationary path in $\text{AdS}_2 \times \mathbf{S}^2$, where the particle stays at a constant radial distance from the boundary determined solely by its physical (equivalently central) charges. **(a)** The effective potential exhibits a minimum at $\psi = \tan^{-1}(-\sqrt{j^2}/q_e)$ that has precisely $V(\psi_{\min}) = 0$. **(b)** Intrinsic trajectory in global (strip) coordinates.

easily verified by inserting (A.15) into eqs. (A.6) and (A.8) or, alternatively, by noticing that the effective radial potential $V(\psi)$ defined in (A.9) develops in this case a global minimum with zero energy precisely at $\psi = \tan^{-1}(-|q_m|/q_e)$, see Figure 9.

Let us remark that, from this perspective, the purely electric (i.e., extremal) and purely magnetic setups are also singled out, since the stationary motion occurs either at the boundary—reflecting the no-force condition experienced by the particle and the black hole, or at the 'middle' of AdS, namely at $\psi = \pi/2$, with all intermediate cases falling in between.

Next, allowing for configurations with non-vanishing angular momentum $\ell$ within $\mathbf{S}^2$ results in the exact same analysis, with suitable minor modifications. For instance, the global energy now reads (cf. (A.16))

$$K_+^2 = j^2 = \ell^2 + q_m^2 \,. \tag{A.19}$$

such that (A.17) becomes

$$\tan \psi = -\frac{\sqrt{j^2}}{q_e} \,, \tag{A.20}$$

hence implying that the equilibrium radius (A.20) is sent towards the 'middle' of $\text{AdS}_2$ as $\ell$ increases. Consequently, and similarly to what happened with the static paths (A.15), the auxiliary radial potential in AdS exhibits in this case a global minimum at $\psi = \tan^{-1}(-|j|/q_e)$ satisfying $V(\psi_{\min}) = 0$. This suggests, in turn, that they could also preserve a subset of the supersymmetries exhibited by the $\text{AdS}_2 \times \mathbf{S}^2$ background. The full proof of this statement appears in [148]. Here, we focus instead on showing that the worldline Hamiltonian associated to (global) time translations satisfies a BPS bound saturated precisely by trajectories of the form (A.12), with $K_0 = 0$ and $K_+ = K_- = \pm j$.

Consider then the action (A.1). Using the global time as worldline parameter, namely $\sigma = \tilde{\tau}$, we have

$$S_{wl} = -\tilde{m} \int d\tilde{\tau} \left[ \sqrt{\csc^2 \psi - \csc^2 \psi \left( \frac{d\psi}{d\tilde{\tau}} \right)^2 - \left( \frac{d\theta}{d\tilde{\tau}} \right)^2 - \sin^2 \theta \left( \frac{d\phi}{d\tilde{\tau}} \right)^2} + \frac{q_e}{\tan \psi} + q_m \cos \theta \frac{d\phi}{d\tilde{\tau}} \right] . \tag{A.21}$$

From here one readily computes the Hamiltonian

$$\mathcal{H} = \mathsf{P}_i \frac{d}{d\tilde{\tau}} \mathsf{q}^i - \mathcal{L} = \csc \psi \sqrt{\tilde{m}^2 + \sin^2 \psi \, \mathsf{P}_\psi^2 + \mathsf{P}_\theta^2 + \csc^2 \theta \left( \mathsf{P}_\phi + q_m \cos \theta \right)^2} + \frac{q_e}{\tan \psi} , \tag{A.22}$$

with the conjugate momenta being

$$\mathsf{P}_\psi = \frac{\tilde{m}}{\sqrt{-h_{\tau\tau}}} \csc^2 \psi \frac{d\psi}{d\tilde{\tau}} , \qquad \mathsf{P}_\theta = \frac{\tilde{m}}{\sqrt{-h_{\tau\tau}}} \frac{d\theta}{d\tilde{\tau}} , \qquad \mathsf{P}_\phi = \frac{\tilde{m}}{\sqrt{-h_{\tau\tau}}} \sin^2 \theta \frac{d\phi}{d\tilde{\tau}} - q_m \cos \theta , \tag{A.23}$$

whilst

$$h_{\tau\tau} = g_{\mu\nu} \frac{dx^\mu}{d\tilde{\tau}} \frac{dx^\nu}{d\tilde{\tau}} = \frac{-\tilde{m}^2 \csc^2 \psi}{\tilde{m}^2 + \sin^2 \psi \, \mathsf{P}_\psi^2 + \mathsf{P}_\theta^2 + \csc^2 \theta \left( \mathsf{P}_\phi + q_m \cos \theta \right)^2} , \tag{A.24}$$

denotes the pull-back of the spacetime metric onto the worldline. Furthermore, using the explicit form of the conserved angular momentum along the 2-sphere

$$\begin{aligned} \mathsf{J}_1 &= -\sin \phi \, \mathsf{P}_\theta - \cot \theta \cos \phi \, \mathsf{P}_\phi - q_m \csc \theta \cos \phi , \\ \mathsf{J}_2 &= \cos \phi \, \mathsf{P}_\theta - \cot \theta \sin \phi \, \mathsf{P}_\phi - q_m \csc \theta \sin \phi , \\ \mathsf{J}_3 &= \mathsf{P}_\phi , \end{aligned} \tag{A.25}$$

the Hamiltonian (A.22) can be written as follows

$$\mathcal{H} = \csc \psi \sqrt{q_e^2 + \sin^2 \psi \, \mathsf{P}_\psi^2 + \mathsf{J}^2} + \frac{q_e}{\tan \psi} . \tag{A.26}$$

Notice that the minimum value for $\mathcal{H}$ occurs when $\mathsf{P}_\psi = 0$ and $\cos \psi = -q_e / \sqrt{q_e^2 + \mathsf{J}^2}$, namely for stationary solutions (in $\text{AdS}_2$) satisfying (A.20). Indeed, we find that for those trajectories the Hamiltonian saturates the following BPS bound [148]

$$\mathcal{H} \geq \sqrt{\mathsf{J}^2} . \tag{A.27}$$

## B  Details on Worldline Instanton Computations

In this appendix, we provide detailed calculations regarding the path integral derivation of certain semiclassical worldline instanton contributions in $\text{AdS}_2 \times \mathbf{S}^2$ (see e.g., [118, 149, 150] for some of the original references). We adopt the same approach as [109], where this computation was carried out without accounting for the motion on the sphere, by approximating the path integral as a zero-dimensional one. Here, we extend their analysis by incorporating the

effect of having non-trivial angular momentum along $\mathbf{S}^2$. These results may be viewed as complementary to the discussion in Section 4.3 of the main text.

Therefore, our aim in what follows will be to compute the one-loop contribution to the quantum effective action due to the wordline instanton solutions of the euclidean theory describing the zero-brane probe dynamics in the near-horizon (extremal) black hole geometry. This amounts to performing the following quantum mechanical path integral

$$
\mathsf{Z}_{1-\text{loop}}[A] = - \int_0^\infty \frac{dh}{h} \int \mathcal{D}t_E(h)\mathcal{D}\rho(h) \ e^{-S_{\text{E}}[x^i, A]} \,, \tag{B.1}
$$

which should be restricted to and summed over arbitrary closed paths lying completely in $\mathbb{H}^2 \times \mathbf{S}^2$.[28] In principle, a full determination of (B.1) may be obtained using the heat kernel method [151], which focuses on the exact diagonalization of the Hamilton operator (4.58) responsible for generating the (euclidean) time translations. For our purposes here it is enough to restrict to the stationary point approximation [114, 115, 152], which computes the leading-order classical term in the path integral together with the one-loop prefactor accounting for all gaussian fluctuations around the saddle. However, instead of considering any possible deformation of the on-shell trajectories first derived in [110] and reviewed in Section 4.3, we will truncate the path integral above to a continuous but finite dimensional family of trajectories including the classical solutions of the equations of motion [109].

Consequently, let us start by rewriting the on-shell paths (4.68) in a more convenient way. In particular, given that the latter describe circles of constant hyperbolic radius $\tanh^{-1}(m_{\text{eff}}/q_e)$, we can use the proper time description to parametrize the trajectories as follows

$$
t_E(\sigma) = t_{E,0} - R_0 \sin\theta(\sigma)\,, \qquad \rho(\sigma) = \rho_0 \mp R_0 \cos\theta(\sigma)\,, \tag{B.2}
$$

where $\theta(\sigma)$ captures the angular motion along the circle. This can be shown to be equal to

$$
\dot\theta = \pm p_{t_E}\rho \implies \theta(\sigma) = 2\tan^{-1}\left( \sqrt{\frac{1 \mp m_{\text{eff}}/q_e}{1 \pm m_{\text{eff}}/q_e}} \tan\left( \frac{\sigma}{2}\sqrt{q_e^2 - m_{\text{eff}}^2} + c_1 \right) \right)\,. \tag{B.3}
$$

Therefore, a useful family of trajectories that incorporates the former classical solutions can be obtained by considering circular paths within $\mathbb{H}^2$ with arbitrary location for their centers $(t_{E,0}, \rho_0)$ and all possible values of the radius $R$—with the obvious constraint of having $R \leq \rho_0$. The paths thus described are of the form

$$
t_E(\sigma) = t_{E,0} - R\sin\theta(\sigma)\,, \qquad \rho(\sigma) = \rho_0 - R\cos\theta(\sigma)\,, \tag{B.4}
$$

with $\dot\theta = q_e\rho/\rho_0$, such that

$$
\theta(\sigma) = 2\tan^{-1}\left( \sqrt{\frac{\rho_0/R - 1}{\rho_0/R + 1}} \tan\left( \frac{q_e\sigma}{2\rho_0}\sqrt{\rho_0^2 - R^2} \right) \right)\,, \tag{B.5}
$$

---

[28]A recent analysis including the spatial dependence in the full black hole background can be found in [122].

and are completely characterized by a finite number of (continuous) parameters $(t_{E,0}, \rho_0, R)$. Notice that one recovers the on-shell solutions upon imposing

$$R = \frac{m_{\text{eff}}}{q_e} \rho_0 \,, \qquad \rho_0 = \frac{q_e}{p_{t_E}} \,, \tag{B.6}$$

which are periodic in proper time with period $T$ equal to $2\pi\rho_0/(q_e\sqrt{\rho_0^2 - R^2})$. Therefore, truncating the path integral (B.1) to this 2-parameter family of off-shell trajectories and using conformal invariance [109], results in the following zero-dimensional QFT

$$\mathsf{Z}_{\text{mini}}[A] = -\int \frac{dt_{E,0}d\rho_0}{\rho_0^2} \int \frac{RdR}{\rho_0^2} \int_0^\infty \frac{dh}{h} \, e^{-S_E} \,. \tag{B.7}$$

On the other hand, the closed instanton action, when evaluated on the paths (B.4), yields

$$\begin{aligned}
S_E &= \int_0^T d\sigma \left[ \frac{(q_e R)^2}{2h\rho_0^2} + \frac{hm_{\text{eff}}^2}{2} - q_e^2 \frac{\rho - \rho_0}{\rho_0} \right] \\
&= \pi \left[ \frac{q_e R^2}{h\rho_0\sqrt{\rho_0^2 - R^2}} + \frac{hm_{\text{eff}}^2\rho_0}{q_e\sqrt{\rho_0^2 - R^2}} \right] - \frac{q_e^2 R}{\rho_0} \int_0^T d\sigma \cos\theta(\sigma) \,,
\end{aligned} \tag{B.8}$$

where to obtain the second equality we used the relations

$$\begin{aligned}
\dot{t}_E^2 + \dot{\rho}^2 &= R^2\dot{\theta}^2 = (q_e R)^2 \frac{\rho^2}{\rho_0^2} \,, \\
\dot{t}_E &= \dot{\theta}(\rho - \rho_0) = \frac{q_e\rho}{\rho_0}(\rho - \rho_0) \,.
\end{aligned} \tag{B.9}$$

Furthermore, noticing that

$$\int_0^T d\sigma \cos\theta(\sigma) = \frac{2\pi\rho_0}{q_e R} \left( \frac{\rho_0}{\sqrt{\rho_0^2 - R^2}} - 1 \right) \,, \tag{B.10}$$

one may rewrite (B.8) as follows

$$S_E = \pi \left[ \frac{q_e R^2}{h\rho_0\sqrt{\rho_0^2 - R^2}} + \frac{hm_{\text{eff}}^2\rho_0}{q_e\sqrt{\rho_0^2 - R^2}} - 2q_e\left( \frac{\rho_0}{\sqrt{\rho_0^2 - R^2}} - 1 \right) \right] \,. \tag{B.11}$$

To show that the stationary points of the truncated path integral indeed correspond to the on-shell trajectories (B.2) one needs to extremize the action (B.11) with respect to both $h$ and $R$. From the einbein condition one finds

$$\frac{\partial S_E}{\partial h} = 0 \implies h_{\min} = \frac{q_e}{m_{\text{eff}}} \frac{R}{\rho_0} \,, \tag{B.12}$$

which when inserted back into the minisuperspace action yields

$$S_E = 2\pi \left[ \frac{m_{\text{eff}}}{\sqrt{\rho_0^2/R^2 - 1}} - q_e\frac{\rho_0/R}{\sqrt{\rho_0^2/R^2 - 1}} + q_e \right] \,. \tag{B.13}$$

Similarly, extremizing (B.13) with respect to $R$ leads to

$$R_{\text{min}} = \frac{m_{\text{eff}}}{q_e}\rho_0 \implies S_{\text{E, min}} = 2\pi\left(q_e - \sqrt{q_e^2 - m_{\text{eff}}^2}\right), \tag{B.14}$$

thus reproducing the result shown in eq. (4.65). More generally, one can determine the first order variation of $S_{\text{E}}$

$$\delta S_{\text{E}} = \pi\left[-\frac{q_e R}{h^2\rho_0\sqrt{\rho_0^2/R^2 - 1}} + \frac{m_{\text{eff}}^2\,\rho_0/R}{q_e\sqrt{\rho_0^2/R^2 - 1}}\right]\delta h$$
$$+ \pi\left[\frac{q_e}{h\rho_0\sqrt{\rho_0^2/R^2 - 1}} + \frac{q_e\,\rho_0}{R^2\left(\rho_0^2/R^2 - 1\right)^{3/2}}\left(h^{-1} - 2 + h\frac{m_{\text{eff}}^2}{q_e^2}\right)\right]\delta R, \tag{B.15}$$

from which the stationary values for $(h, R)$ readily follow

$$h_{\text{min}} = \frac{m_{\text{eff}}}{q_e}\rho_0, \qquad \frac{R_{\text{min}}}{\rho_0} = \sqrt{2 - 2h + \frac{h^2 m_{\text{eff}}^2}{q_e^2}}. \tag{B.16}$$

Next, one may compute the second order variation of the instanton action

$$\delta^2 S_{\text{E}} = \frac{\delta^2 S_{\text{E}}}{\delta h^2}(\delta h)^2 + 2\frac{\delta^2 S_{\text{E}}}{\delta h \delta R}\delta h \delta R + \frac{\delta^2 S_{\text{E}}}{\delta R^2}(\delta R)^2, \tag{B.17}$$

with

$$\frac{\delta^2 S_{\text{E}}}{\delta h^2} = \frac{2\pi q_e R}{h^3\rho_0\sqrt{\rho_0^2/R^2 - 1}} \overset{(\text{B.16})}{=} \frac{2\pi m_{\text{eff}}}{\sqrt{q_e^2/m_{\text{eff}}^2 - 1}},$$

$$\frac{\delta^2 S_{\text{E}}}{\delta h \delta R} = -\frac{\pi q_e}{h^2\rho_0\sqrt{\rho_0^2/R^2 - 1}} - \frac{\pi q_e\rho_0}{h^2 R^2\left(\rho_0^2/R^2 - 1\right)^{3/2}} + \frac{\pi h m_{\text{eff}}^2 R}{q_e\rho_0^2\left(1 - R^2/\rho_0^2\right)^{3/2}} \overset{(\text{B.16})}{=} -\frac{2\pi q_e}{\rho_0\sqrt{q_e^2/m_{\text{eff}}^2 - 1}},$$

$$\frac{\delta^2 S_{\text{E}}}{\delta R^2} = -\frac{2\pi q_e\rho_0\left((2h)^{-1} - 2 + h\frac{m_{\text{eff}}^2}{q_e^2}\right)}{R^3\left(\rho_0^2/R^2 - 1\right)^{3/2}} + \frac{3\pi q_e\rho_0^3\left(h^{-1} - 2 + h\frac{m_{\text{eff}}^2}{q_e^2}\right)}{R^5\left(\rho_0^2/R^2 - 1\right)^{5/2}} \overset{(\text{B.16})}{=} -\frac{2\pi q_e^2}{m_{\text{eff}}\rho_0^2\left(q_e^2/m_{\text{eff}}^2 - 1\right)^{3/2}}. \tag{B.18}$$

Using these results, we can approximate (B.7) at one-loop order by a sum over the different instanton sectors[29]

$$\mathsf{Z}_{\text{mini}}[A] = -\sum_{\text{saddles}}\int\frac{dt_{E,0}d\rho_0}{\rho_0^2}\frac{R_{\text{min}}}{h_{\text{min}}\rho_0^2}\mathcal{A}_{1-\text{loop}}\,e^{-S_{\text{E, min}}} + (\text{higher-order})$$

$$= i\int\frac{dt_{E,0}d\rho_0}{\rho_0^2}\sum_{k=1}^{\infty}\sum_{\ell=0}^{\ell_{\text{max}}}\frac{\left(q_e^2 - \tilde{m}^2 - \ell(\ell+1)\right)}{2\pi k q_e^3}e^{-2\pi k\left(q_e - \sqrt{q_e^2 - \tilde{m}^2 - \ell(\ell+1)}\right)} + (\text{higher-order}), \tag{B.19}$$

---

[29] Recall that for fixed extremality parameter $q_e/\tilde{m}$ there is a maximum angular momentum for which an instanton mediating the non-perturbative decay can exist, since $\ell(\ell+1) \leq q_e^2 - \tilde{m}^2$ must be satisfied.

where in the second step we inserted the fluctuations around each saddle point, which read

$$\mathcal{A}_{1-\text{loop}} = \left( \det \frac{\delta^2 S_{\text{E}}}{\delta x^i \delta x^j} \right)^{-1/2} = \frac{m_{\text{eff}} \rho_0 \left( q_e^2/m_{\text{eff}}^2 - 1 \right)}{2\pi i q_e^2} \,. \tag{B.20}$$

Notice the imaginary phase, which arises from the fact that the determinant of the Hessian is negative definite and therefore accounts for the instability of the vacuum via Schwinger pair production.

Before closing this appendix, let us make a couple of comments concerning eq. (B.19). First, one can easily check that the $\ell = 0$ sector reproduces the result obtained in [109]. However, a proper treatment of the problem would require to sum over all possible angular momenta, according to the fact that the calculation is actually four-dimensional. Second, we see that the one-loop prefactor is such that the contribution turns off in the extremal case, namely when $|q_e| = m_{\text{eff}}$. This seemingly innocuous observation plays a crucial role in our discussion in Section 4.3, since it ultimately explains the absence of non-perturbative corrections to the D0-D2-D4 black hole entropy due to D0-brane states, as shown in [26]. Finally, we should remark that even if the computation carried out herein is only partial, a more rigorous approach including the effect of all possible off-shell path deformations around the stationary point [122], or rather by computing the vacuum persistence amplitude via scattering methods [118] or direct quantum integration [99], confirms both the quantitative dependence of the instanton actions here determined as well as the qualitative behavior of the exact prefactor appearing in (B.20).

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
