# Peer review of "Quantum Calabi-Yau Black Holes and Non-Perturbative D0-brane Effects"

_SciPost Physics_

## Round 1 · Referee Report · Anonymous (Referee 1) · 2025-9-26

Report

This paper provides new insights into non-perturbative effects in string theory type-IIA compactifications in the large volume limit. It builds upon Ref. [26] which observed that D0-branes do not give rise to non-perturbative corrections of black hole entropy, if the the black hole carries either purely electric or purely magnetic D0-charge.

The paper shows that for black holes with a generic charge configuration the black hole entropy is corrected by non-perturbative D0-brane effects, and it gives a good and intuitive argument why these corrections vanish for purely electric can purely magnetic charges.

Both results rely on extending the work of Ref [38] to include higher-derivative corrections. In order to express the black hole entropy in terms of charges, the so-called attractor equations have to be solved. Even in the large volume limit, this leads to systems of coupled algebraic equations which cannot be solved in closed form in general. The authors manage to find, in the large volume limit, a solution which is almost explicit, except that one quantity can only be determined iteratively. This is an interesting and useful result, and a substantial extension of previous work, including Ref. [38]. The authors also make useful observations about the utility of a subset of symplectic transformations as solution-generating transformations; this allows them to use simpler charge configurations in their arguments.

Based on these results, they use the Schwinger-type representation of perturbative and non-perturbative corrections to the (generalized) prepotential resulting from the relation between IIA string theory and M-theory, to show that for generic charges the prepotential and black hole entropy receive non-perturbative corrections.

The rest of the paper provides a simplified analysis of why the corrections vanish for black holes which carry purely electric or purely magnetic charges. The authors compute the trajectories of probe D0-branes in the near-horizon geometry, showing that the geodesic equation is integrable. Using semi-classical analysis, they show that for electric charge any non-perturbative effect will have a zero pre-factor, while in the magnetic case non-perturbative effects cannot arise in the first place. As the authors point out, there is room to extend this to a full quantum analysis as well as getting a more complete understanding of the non-perturbative effects for generic charge configurations. But for the purpose of understanding why purely electric and purely magnetic D0 charge is special, their simplified argument is convincing, and has the advantage of being intuitive and being consistent with arguments from other work.

Overall, this paper makes a substantial contribution to an interesting and important topic. Regarding criteria for publication, this paper opens up several pathways for further research, while solving the problems raised by the results of Ref [26]. I am happy to recommend this paper for publication, and have a few comments that I like the authors to consider (see requested changes).

Requested changes

1.Below (2.1) it is stated that the hypermultiplet manifold is parametrized by complex coordinates. However, a general quaternionic-Kaehler manifold need not have complex structure. (This is not relevant for anything following, but still sounded odd to me.)

  1. In (2.8) a different symbol $\delta \mathcal{F}_g^{\mathrm{hyp}}$ is used for what was previously denoted $\mathcal{F}_g$. Or is there a difference?

  2. In the context of equation (3.51) it is explained in detail that the first equality states that the entropy is a symplectic invariant. It may be worth adding that the equality between the first and third expression is the statement that it is a symplectic function. (There is no explicit definition of symplectic function in the text - while I agree that every reader will know how a function/scalar field transforms, there is a non-zero chance of confusion which can be avoided by being explicit).

  3. Between (4.11) and (4.12) the distinction between local and global $AdS_2 \times S^2$ is stressed. The text can be read as suggesting that the near horizon geometry is globally $AdS_2 \times S^2$, but I don't think that this is the case. Is the distinction between local and global $AdS_2 \times S^2$ relevant for the following arguments, or is it just convenient to use global coordinates while the arguments are valid when just using a sufficiently large but not necessarily geodesically complete part of global $AdS_2 \times S^2$? I don't think this is critical but suggest to clarify this point.

  4. In footnote 18, is the second name really Swanziger (not Zwanziger)?

Recommendation

Publish (easily meets expectations and criteria for this Journal; among top 50%)

---

## Round 1 · Referee Report · Anonymous (Referee 2) · 2025-9-30

Report

This paper discusses BPS black holes in Calabi–Yau compactifications of Type IIA string theory and focuses on the perturbative and non-perturbative corrections to their entropy. The paper is a follow-up from a previous work [26], where it was shown that the perturbative corrections can be resummed in the large volume regime, which revealed potential non-perturbative contributions to the BH entropy. It turned out that these new corrections did not affect the D0-D2-D4 and D2-D6 black holes in [26]; the aim of this paper is to understand what black holes are affected by them and what is their physical interpretation.

In the first part of the paper, after reviewing the 4d N=2 setup in the presence of the infinite tower of BPS higher-derivative corrections, the authors sketch how these corrections can be resummed using IIA/M-theory duality. They then discuss the BPS attractor mechanism in the large volume regime in the presence of such corrections, and derive an explicit formula for the quantum-corrected entropy of a general BH with D0-D2-D4-D6 charges. This result significantly generalizes previous results in the literature, by allowing for all charges and including corrections. They show that their general formula correctly reproduces the previous results in the literature, and then discuss how one can use symplectic rotations to generate other BH solutions starting from one. Through this they show that the D0-D2-D4 and D2-D6 BHs are very special cases, and that non-perturbative corrections from D0-branes will generically correct the entropy of a BH with more general charges.

In the second part of the paper they seek to understand the physical origin of these non-perturbative corrections, and why they do not affect the D0-D2-D4 and D2-D6 BHs. They want to interpret them as coming from the presence of light D0 branes in the large volume limit. To argue for this, they perform a semi-classical analysis of a charged particle in the near-horizon region of a (double) extremal classical black hole. They are able to show that the interaction between the probe particle and the black hole is determined by a relative phase between the central charges of the BH and the particle. They show that for a general black hole D0 branes become effectively sub-extremal near the horizon and fall inside the black hole. Furthermore it allows them to single out the special cases of the D0-D2-D4 and D2-D6 BHs and explain why the D0 corrections do not affect them (they are seen as purely electric / magnetic from the perspective of a D0 probe particle). The picture is simplified but convincing, and they point to future work where quantum effects and the back-reaction will be taken into account.

This paper is interesting and timely, and provides ideas and formulas that will lead to future publications. I am happy to recommend this for publication, once the minor points below are addressed.

Requested changes

1- It is conjectured that non-perturbative corrections are needed to reproduce microscopic degeneracies. The authors then say they focus on a specific type of such corrections (from the D0 branes becoming light). The paper reads as though these corrections recover the fully corrected BH entropy (that matches the microscopic counting), is this the case (at least for these types of BHs that are in the large volume limit))?

2- To clarify, does "taking the $p^0 \to 0$ limit" just mean being careful about setting $p^0=0$ (because it is naively singular, unlike setting $p^a=q_0=0$) ? At no point should the authors be considering BHs with $p^0$ charge arbitrarily close to zero but non-zero.

3- While the computations in Section 4 are detailed and clear, I was confused about when the particle probe under consideration should be thought of as a general charged particle in 4d N=2 or an actual D0-particle. The lightest particle states in this limit are D0 states, but the others also become light, so perhaps it does not matter for most of the discussion? In Figure 6, things are clear, but perhaps it would be worth emphasizing the difference earlier in Section 4.

Typos:

4- Typo in ref [76]: superrgravity

5- Page 1 - “has led to conjecture” is missing a subject or a "the"

Recommendation

Publish (easily meets expectations and criteria for this Journal; among top 50%)

---

## Editorial Decision

awaiting_resubmission